# FKRP-dependent glycosylation of fibronectin regulates muscle pathology in muscular dystrophy

A. J. Wood[1,2], C. H. Lin [3,4], M. Li[1,2], K. Nishtala [3,4], S. Alaei[1,2], F. Rossello[1,5], C. Sonntag[1,2], L. Hersey[1,2], L. B. Miles [1,6], C. Krisp[3], S. Dudczig[1,2,7], A. J. Fulcher [8], S. Gibertini[11], P. J. Conroy [9,10], A. Siegel[1,2], M. Mora [11], P. Jusuf[1,2,7], N. H. Packer [3,4] & P. D. Currie [1,2✉]

The muscular dystrophies encompass a broad range of pathologies with varied clinical outcomes. In the case of patients carrying defects in fukutin-related protein (FKRP), these diverse pathologies arise from mutations within the same gene. This is surprising as FKRP is a glycosyltransferase, whose only identified function is to transfer ribitol-5-phosphate to α-dystroglycan (α-DG). Although this modification is critical for extracellular matrix attachment, α-DG's glycosylation status relates poorly to disease severity, suggesting the existence of unidentified FKRP targets. Here we reveal that FKRP directs sialylation of fibronectin, a process essential for collagen recruitment to the muscle basement membrane. Thus, our results reveal that FKRP simultaneously regulates the two major muscle-ECM linkages essential for fibre survival, and establishes a new disease axis for the muscular dystrophies.

[1] Australian Regenerative Medicine Institute (ARMI), Faculty of Medicine, Nursing and Health Science, Monash University, Clayton, VIC, Australia. [2] EMBL Australia, Victorian Node, Faculty of Medicine, Nursing and Health Science, Monash University, Clayton, VIC, Australia. [3] Biomolecular Discovery and Design Research Centre, Department of Molecular Sciences, Faculty of Science & Engineering, Macquarie University, Sydney, NSW, Australia. [4] Institute for Glycomics, Griffith University, Gold Coast, QLD, Australia. [5] University of Melbourne Centre for Cancer Research, The University of Melbourne, Melbourne, VIC, Australia. [6] School of Biological Sciences, Monash University, Clayton, VIC, Australia. [7] School of Bio-Sciences, Faculty of Science, The University of Melbourne, Parkville, VIC, Australia. [8] Monash Micro Imaging Facility, Monash University, Clayton, VIC, Australia. [9] Department of Biochemistry and Molecular Biology, Faculty of Medicine, Nursing and Health Science, Monash University, Melbourne, VIC, Australia. [10] Monash Antibody Technologies Facility (MATF), Monash University, Clayton, VIC, Australia. [11] Muscle Cell Biology Laboratory, Neuromuscular Diseases and Neuroimmunology Unit, Fondazione IRCCS Istituto Neurologico "Carlo. Besta" Via Temolo 4, Milano, Italy. ✉email: peter.currie@monash.edu

The dystroglycanopathies are amongst the most enigmatic of all muscular dystrophies (MDs)[1–3], largely due to heterogeneity of distinct clinical presentations that remain unexplained at a mechanistic level[4–7]. Within this class of disorders, mutations in *FKRP* result in a particularly varied pathological spectrum, with the presence of disease modifiers and novel modes of FKRP action, both being put forward as possible explanations for this phenotypic variation[1,8]. Furthermore, the glycosylation status of dystroglycan, the only known target of FKRP activity, corresponds poorly to phenotype severity[9], an observation that suggests the possibility of dystroglycan-independent modes of FKRP function.

## Results

**fkrp mutants exhibit a loss of muscle integrity and function.** In order to better understand the basis for this diversity of phenotypic severity, we developed zebrafish loss-of-function models of FKRP deficiency using distinct genome editing strategies (Extended Data Fig. 1). The *fkrp* mutant fish generated by these different processes possessed similar phenotypes and were shown to generate nonsense-mediated decay of *fkrp* transcripts and a consequential loss of protein by western blot (Extended Data Fig. 1d, e). Furthermore, *fkrp* mutant larvae possessed a significant loss of (IIH6) reactivity, a marker of glycosylated dystroglycan, FKRP's canonical target. This

loss of immunoreactivity could not be attributed to an overall loss of dystroglycan protein, or loss of dystroglycan's core-binding partner, Laminin, whose deposition was actually significantly increased at the muscle basement membrane in homozygous mutant *fkrp* larvae (Extended Data Fig. 2). We, therefore, concluded that the mutations we have generated resulted in the loss of FKRP function. Consequently, we reasoned that by directly comparing the phenotype of our newly generated *fkrp⁻ᐟ⁻* alleles (Extended Data Figs. 1–3) to that which presents in *α-dg* (*dag1⁻ᐟ⁻*) mutants, the canonical target of FKRP glycosylation[10], novel modes of FKRP function could be identified.

To perform these analyses, the phenotypes of *fkrp⁻ᐟ⁻* and *dag1⁻ᐟ⁻* mutants were compared for muscle structure and integrity defects at 3 and 5 days post fertilisation (dpf), a period during which progressive muscular dystrophy and muscle basement membrane (MBM) attachment defects were evident in our previously generated zebrafish models of muscular dystrophy[11–13]. Both models exhibited the expected fibre detachment pathology over this period, the prevalence of which increased with age and *fkrp* mutants also demonstrated Evan's blue dye (EBD) uptake into muscle fibres, which provides evidence for sarcolemmal defects in these mutants (Fig. 1a–c and Extended Data Figs. 2d, f–f'). Mutants also exhibited a phenotypically proportional loss of maximal active force generation (Fig. 1e)[14]. Furthermore, both *dag1* and *fkrp* mutants possess highly similar retinal basement membrane defects that reflect dystrophin

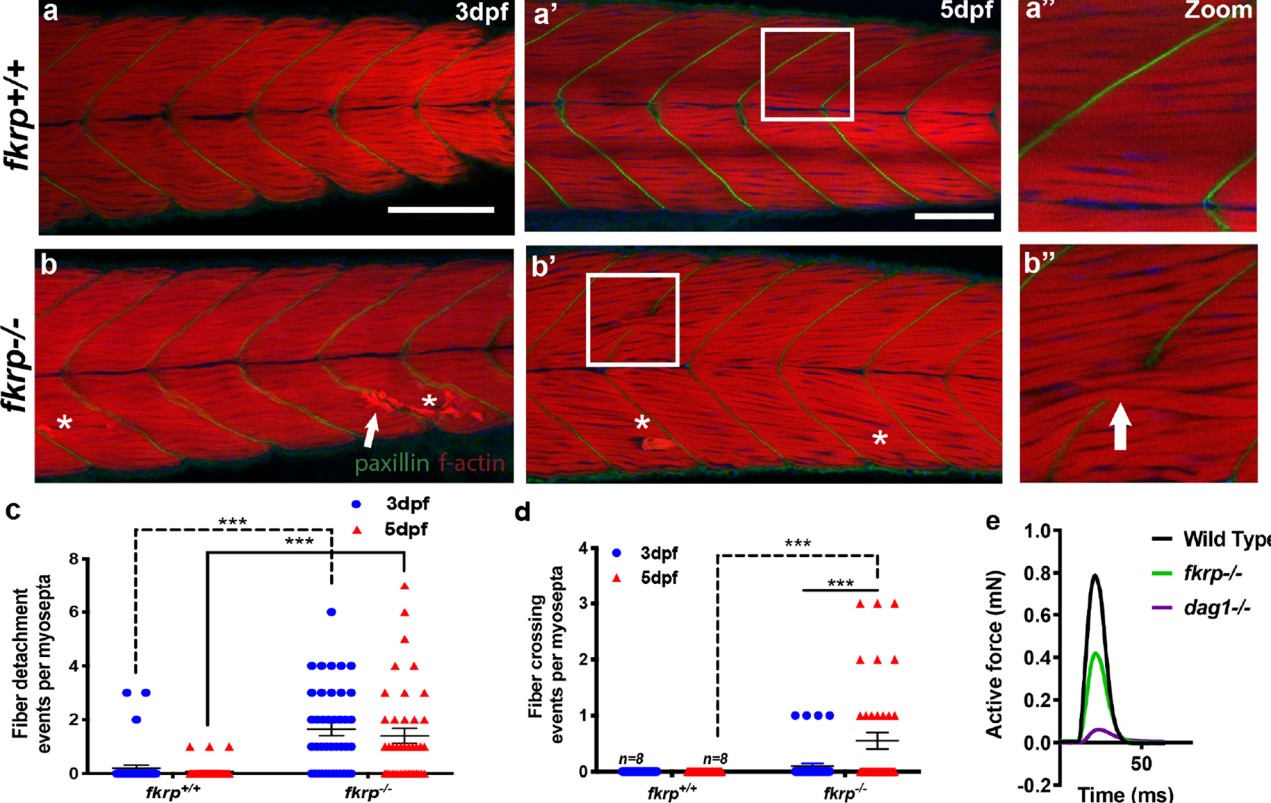

**Fig. 1 fkrp⁻ᐟ⁻ fish possess a basement membrane failure phenotype. a–d** Muscle fibre-crossing and detachment phenotypes at the muscle basement membrane of *fkrp⁻ᐟ⁻* mutant larvae. **a–b″** Z-projected images from the entire mediolateral extent of the myotome centred on the region of the anal pore. Phalloidin conjugated to TRITC (red) stains for f-actin as a marker of skeletal muscle and paxillin (green) visualises vertical myosepta muscle basement membrane integrity. White arrowheads: muscle fibres crossing at defective muscle basement membranes, asterisks: muscle fibre detachment. 3 dpf, scale bar = 100 µm. 5 dpf, scale bar = 75 µm, white boxes in **a′**, **b′** denote the high-magnification views presented in **a″** and **b″**. **c, d** Quantitation of fibre detachment (**c**) and basement membrane crossing events (**d**) per myosepta in wild-type (n = 9) or *fkrp* mutant larvae (n = 9) over five independent experiments, analysed using two-way ANOVA multiple comparisons assuming non-parametric data, the significance of ***(P = < 0.0001), error bars represent SEM. **e** Physiological analysis of muscle function. The maximum active force (mN) generated over a specific time interval (ms) was measured in individual genotypes of 6 dpf larvae represented in single-twitch recordings of *fkrp⁺ᐟ⁺* sibling (sib) black line (n = 8), *fkrp⁻ᐟ⁻* green line (n = 6) and *dag1⁻ᐟ⁻* purple line (n = 5).

glycoprotein complex (DGC) component deposition deficits that accompany a lack of α-dg-mediated attachment (Extended Data Fig. 3). The similar fibre detachment and retinal basement membrane phenotypes of these mutants are consistent with the failure of FKRP-dependent α-dg glycosylation, resulting in a loss of cellular attachment in these different contexts. We could provide no evidence for the previously reported neuronal or vascular defects in *fkrp* morpholino-injected fish by staining for neuronal synapses, fluorescent-dextran-based angiography or EBD vascular injection, which revealed only muscle-related uptake within damaged fibres (Extended Data Fig. 2f–g')[13,15].

**Muscle basement membrane defects in *fkrp* mutants.** Intriguingly, however, *fkrp*[−/−] mutant larvae also possessed a specific phenotype, not evident in *dag1*[−/−] mutants. *fkrp* homozygous mutants exhibited muscle fibres that crossed or bypassed attachment to the MBM of the vertical myosepta, a phenotype indicative of MBM failure (Fig. 1a, b, d and Extended Data Fig. 1i). As the identical fibre-crossing phenotype had been observed previously in *fibronectin* loss-of-function larvae[16], we postulated that FKRP may play a role in regulating fibronectin function. Fibronectin presented as an attractive candidate for a FKRP target protein as it is highly decorated with *N*- and *O*-linked glycans and plays a central role in forming and maintaining the MBM[16–18]. Furthermore, one of the principal roles of fibronectin is to recruit collagen to stabilise BM formation. This biological function is a fundamental axis of tissue integrity in many organ systems, critical both to tissue homoeostasis and to diseases such as cancer, where failure in BM cell adhesion plays a key role in the transition to metastasis[19]. To determine if disruption of collagen recruitment and consequent MBM formation could explain the phenotypes we observed, *fkrp*[−/−], *dag1*[−/−] and wild-type control fish were immunohistochemically stained for fibronectin at 1 dpf and for collagen-1a at 5 dpf, which are the respective time points at which fibronectin and collagen accumulation can be visualised and quantitated. Fibronectin localised correctly to the MBM in mutant and control fish (Fig. 2a, e) but unexpectedly exhibited a significantly higher ($P < 0.0001$) deposition in the *fkrp*[−/−] fish than controls (Fig. 2e). Collagen-1 protein deposition, in contrast, was dramatically and significantly reduced ($P < 0.0001$) in *fkrp*[−/−] mutants compared to control and *dag1*[−/−] larvae (Fig. 2b, f) at the MBM. Next, transmission electron microscopy (TEM) was used to ascertain collagen deposition between the MBM of the zebrafish vertical myosepta in *fkrp*[−/−] ($n = 9$), *dag*[−/−] ($n = 7$) and wild-type controls ($n = 8$) at 7 dpf. At this stage of development, electron-dense collagen fibrils align between adjacent MBMs at the vertical myosepta, providing structural integrity[20]. As expected, *dag1*[−/−] mutants exhibited normal localisation of fibrillar collagen at the MBM, however, in *fkrp*[−/−] mutants, no fibrillar collagen could be detected, confirming the immunohistochemistry-based findings (Fig. 2c, d). To determine if collagen loss in *fkrp*[−/−] fish resulted from a lack of gene expression, transcript levels of *collagen-1a*, the main collagen expressed at the MBM[20], was investigated. However, neither *collagen-1a* nor *fibronectin* gene expression levels were reduced in *fkrp*[−/−] mutants. In fact, mRNA levels for *collagen-1a*, were found to be significantly increased ($P < 0.05$) in *fkrp*[−/−] mutants (Extended Data Fig. 4), indicating that a lack of gene expression is not the cause of the loss of collagen-1a at the MBM of *fkrp*[−/−] mutants (Extended Data Fig. 4a). Next, we sought to determine the physiological consequence for the loss of collagen deposition evident at the MBM of *fkrp*[−/−] mutants. Since collagen is the major ECM constituent that directs passive resistance to strain at the MBM[21], we developed an assay to measure muscle passive force. Our results would predict that *fkrp*[−/−] mutants would exhibit severe reductions in passive force transmission within the larval myotome. In line with this premise, subjecting larval zebrafish to a newly developed passive force assay[22] revealed that *fkrp*[−/−] mutants exhibited a severe and highly significant reduction in their ability to transmit passive force (Fig. 2g), a reduction not evident in either *dag1*[−/−] mutants nor wild-type sibling controls (Fig. 2g).

**Altered fibronectin glycosylation in *FKRP* patient myoblasts.** In order to determine the disease relevance of our observations in the *fkrp* mutant zebrafish model, the role of human fibronectin impairment in muscle cell pathology was investigated in patient-derived FKRP-deficient primary myoblast cells (Table 1 and Extended Data Fig. 4b). To undertake these analyses, two distinct *FKRP*-deficient patient myoblast cell lines from either end of the *FKRP* clinical spectrum were differentiated to form myotubes: those derived from a patient with LGMD2I, a disease with a milder clinical presentation, and cells derived from a patient with a severe CMD. These cells were compared to identically established control lines derived from healthy muscle biopsies (Fig. 3a–d). As FKRP is known to catalyse the transfer of ribitol-5-phosphate to *O*-mannosyl glycan chains on α-DG, we investigated if any glycosylation changes between fibronectin isolated from control and FKRP mutant cells could be identified. Fibronectin from human patient cell lysate was immunoprecipitated and glycans were released and analysed by porous graphitised carbon–liquid chromatography–mass spectroscopy PGC–LC–MS/MS (Fig. 3a)[23]. An analysis of *O*-glycans did not reveal any trace of altered *O*-mannosyl glycans on fibronectin nor changes of mucin-type *O*-glycans between wild-type and control cells (Extended Data Fig. 5). Unexpectedly, however, we detected a highly significant and specific reduction of *N*-glycan sialylation on fibronectin derived from LGMD2I and CMD patient cells when compared to control samples. Specifically, two peaks, *m/z* at 965.9 and 1111.4, corresponding to biantennary, sialylated *N*-glycans with compositions of $NeuAc_1Hex_5HexNAc_4$ and $NeuAc_2Hex_5HexNAc_4$, respectively, were significantly reduced on fibronectin in patient cells when compared to the healthy controls (Fig. 3a, b). In contrast, no general loss of sialylation was detected in the *N*-glycans derived from total cell proteins of healthy control and patient samples (Extended Data Fig. 6), indicating that this decrease in sialylation is specific for fibronectin glycosylation rather than representing a global deficit in protein sialylation in patient cells.

Our analysis thus determined that fibronectin *N*-glycans exhibited a similarly dramatic decrease in sialylation in cells derived from patients that spanned the FKRP clinical spectrum. As a cluster of these biantennary glycans resides within the collagen-binding domain of fibronectin[17], we next determined if a defect in collagen–fibronectin binding could be detected in patient samples, and if this defect scaled with clinical severity. To undertake these analyses, fibronectin was immunoprecipitated and subject to quantitative proteomic analysis to determine the relative abundance of co-precipitated collagen[24]. This analysis revealed that the amount of collagen that co-immunoprecipitated with fibronectin was significantly less in both patient cell lines when compared to healthy controls, suggesting a loss of interaction between fibronectin and collagen, the level of which tracked with clinical severity of the patient cells from which protein was derived (Fig. 3c). To directly assay and quantitate fibronectin–collagen binding in vitro, we undertook surface plasma resonance analyses. Using this method, the binding of native or desialylated fibronectin to immobilised collagen was assessed (Extended Data Fig. 7). This study revealed that the binding affinity of desialylated fibronectin to collagen was

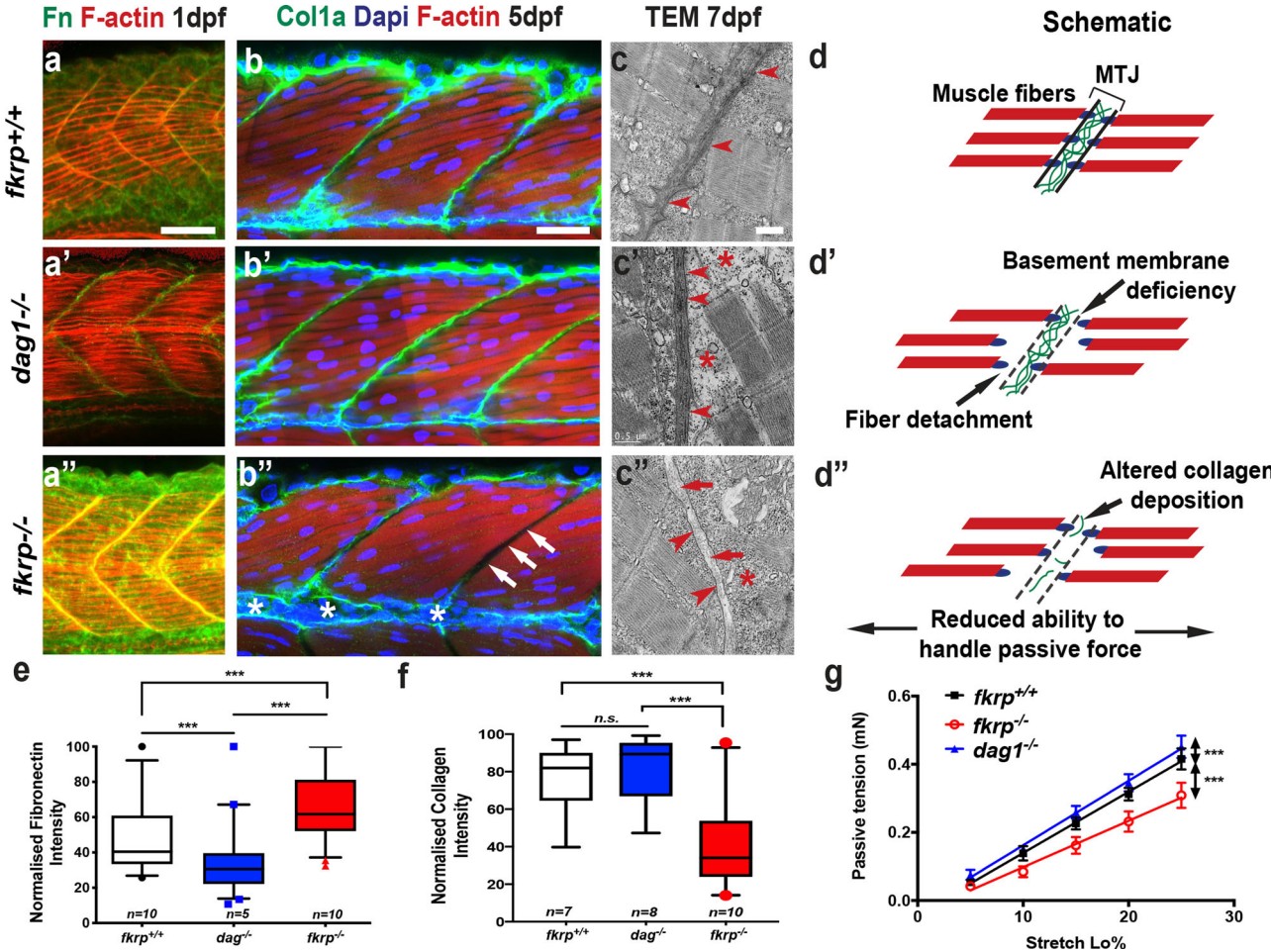

**Fig. 2 The fibronectin–collagen axis is disrupted at the muscle basement membrane in *fkrp* mutants. a–b″** Fibronectin and collagen immunochemistry. Whole-mount staining and confocal imaging of zebrafish myotomes centred on myotome 10 (1 dpf) and myotome 12 (5 dpf). Images are z projections of the mediolateral extent of the myotome stained for f-actin (red) to mark muscle fibres and DAPI (blue) for nuclei. **a** Fibronectin (Fn) staining (green) at the vertical myosepta at 1 dpf, images representative of $n = 9$, three larvae from three separate clutches on different weeks, most severe and weakest phenotypes excluded (scale bar = 30 μm). **a′** $dag1^{-/-}$, **a″** $fkrp^{-/-}$. **b** Collagen-1 (Col1a, green) staining of the muscle basement membrane in wild-type sibling fish. **b′** $dag1^{-/-}$ and **b″** $fkrp^{-/-}$, white arrow: absent collagen, scale bar = 40 μm. **a–b″** Lateral views anterior to the left. **c** Transmission electron micrographs (TEM) of longitudinal sections of 7 dpf zebrafish myotome centred on the muscle basement membrane (red arrowheads). Red arrows: absent collagen fibrils, red star: fibre detachment from vertical myosepta, scale bar = 0.5 μm. Wild-type siblings, **c′** $dag1^{-/-}$ **c″**, $fkrp^{-/-}$, micrographs representative of ($n = 9$), three larvae from three separate clutches on different weeks, most severe and weakest phenotypes excluded. **d** Schematic of fibre-crossing and detachment model at the Myotendinous Junction (MTJ) and disease progression, **d′** canonical DGC axis in $dag1^{-/-}$ and, **d″** combined detachment crossing in $fkrp^{-/-}$. **e, f** Measurement of max intensity from fibronectin (**e**) and collagen (**f**) staining analysed by two-way ANOVA, ***($P = $ <0.0001), plotted points outside 95% confidence interval, box represents 5–95%, median centre line, whiskers = SEM, Tukey's multiple comparison analysis, three independent experiments. **g** Passive force measurement at 6 dpf, calculated by plotting the maximum active force at given loads, as passive tension (mN) against external stretch (Lo%), with representative linear regression analysis of the plotted points of $fkrp^{+/+}$ sibling (sib); black line ($n = 8$), $fkrp^{-/-}$, red line ($n = 6$) and $dag1^{-/-}$ blue line ($n = 5$) was used to test the significance of the differences (***$P = $ <0.0001), error bars = SEM.

**Table 1 Clinical histological phenotype of patient myoblast samples.**

| LGMD2I | CMD |
|---|---|
| Well organised muscle tissue with normotrophic fibres and subsarcolemmal nuclei. Presence of some hypotrophic fibres. No degenerative aspects are observed. No pathological accumulation of PAS positive material. Predominance of type I fibre. Relative hypotrophy of type II fibres. | Severe alteration of the tissue organisation due to abundant peri and endomysial connective tissue, fibre anisometry with normo-, hypo-, atrophic and various hypertrophic fibres some of which with splittings, central nuclei and rare degenerate fibres. |
| Conclusion: Modest, non-specific, myogenic changes. | Conclusion: Morphological features indicative of a severe dystrophic degeneration, compatible with the diagnosis of congenital muscular dystrophy |

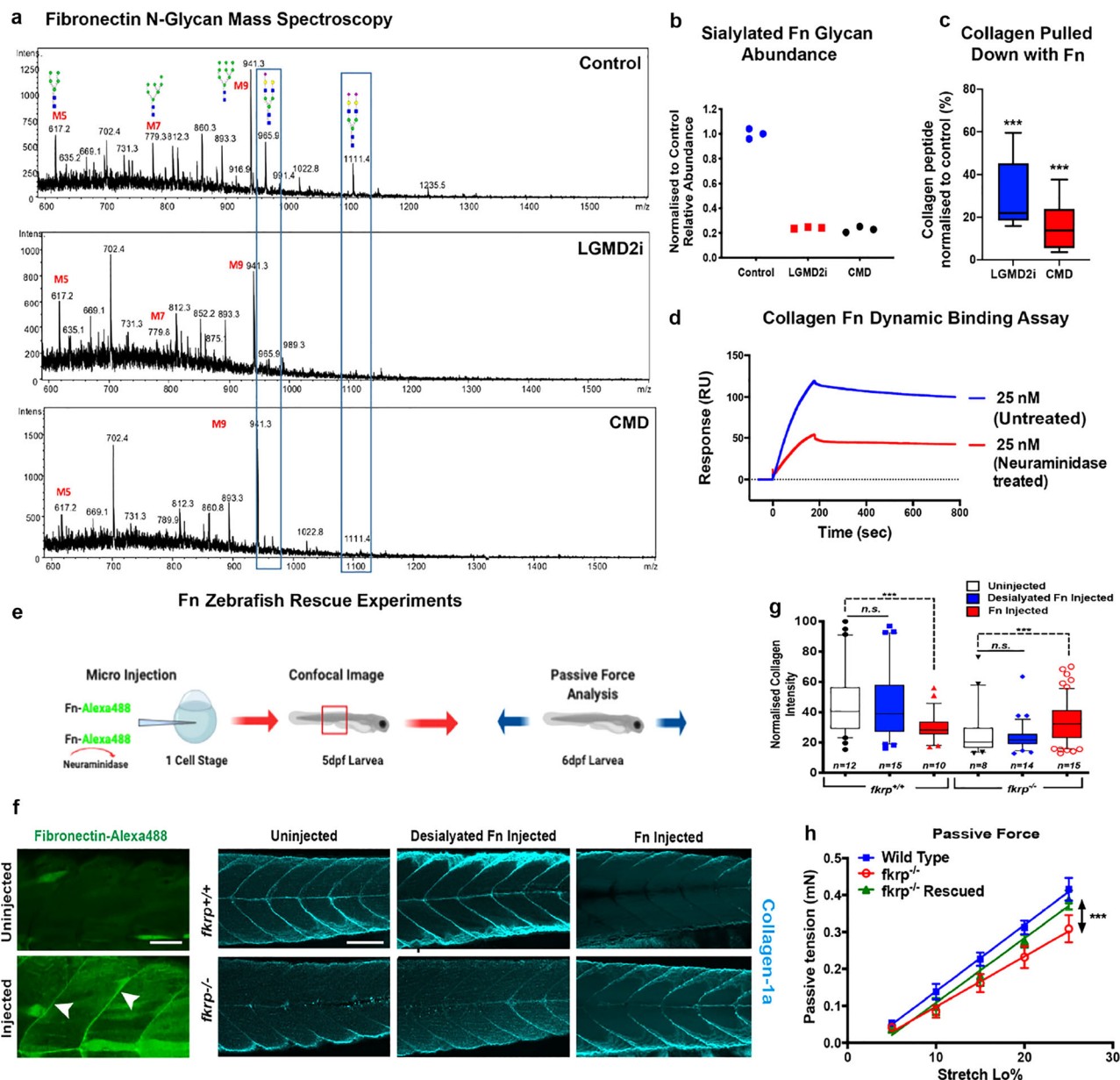

**Fig. 3 FKRP-dependent N-linked sialylation of fibronectin is required for collagen binding and muscle basement membrane localisation in vivo. a** *N*-glycan analysis of muscle fibronectin (Fn) isolated from control and LGMD2I and CMD patient cells. Blue box highlights the area of change at *m/z* 965.9 and *m/z*1111.4. **b** Quantification *n* = 3 averaged from three experimental replicates, each repeated with three technical replicates (*$P < 0.05$). **c** Co-immunoprecipitation of collagen released from purified fibronectin, normalised to healthy human controls (***$P < 0.0001$; three technical repeats repeated three times, box represents 5–95%, median centre line, whiskers = SEM. **d** Collagen and fibronectin binding quantitated via Biacore, before and after treatment with neuraminidase, an enzyme that specifically removes terminal sialic acid residues. **e** Schematic of the fibronectin rescue experiment. **f** Injection of human fibronectin into the single-cell stage of 1-dpf embryos. Fibronectin tagged with Alexa488 localises correctly to the myosepta in wild-type larvae, white arrows. Scale bar = 100 μm. Only fully sialylated fibronectin can rescue collagen deficits evident in *fkrp*$^{-/-}$ larvae. *fkrp*$^{+/+}$ and *fkrp*$^{-/-}$ larvae uninjected or injected, with either desialylated fibronectin or fully sialylated fibronectin, visualised for collagen localisation via z projections of the entire mediolateral extent of the myotome centred on the anal pore. Note also that *fkrp*$^{+/+}$ control fish, injected with sialylated fibronectin, exhibited a lower collagen deposition at the myoseptal boundaries. This lower collagen deposition is attributed to a previously described fibronectin–collagen feedback loop, scale bar = 100 μm. **g** Quantitation of the relative max intensity of collagen deposition at muscle basement membranes, 5 dpf, (***$P < 0.0001$), plotted points outside 95% confidence interval analysed by two-way ANOVA, Dunn's multiple comparisons, from five repeats, box represents 5–95%, median centre line, whiskers = SEM. **h** Physiological rescue after fibronectin injection. Force transducer measurement of passive tension over stretch load % (*n* = 5 for each experimental condition) (***$P < 0.0001$), error bars = SEM.

significantly reduced compared to the fully sialylated native fibronectin (Fig. 3d), suggesting that the decrease in fibronectin sialylation may be causative of the collagen deposition deficits evident in *fkrp*$^{-/-}$ mutants. To further test this hypothesis, fully sialylated or desialylated fibronectin was injected into homozygous *fkrp* mutants, with both forms of fibronectin being tagged with Alexa488 dye to track localisation upon injection. In wild-type and mutant larvae, both native and desialylated fibronectin localised correctly to MBM boundaries (Fig. 3e). Next, we examined collagen expression at the myoseptal boundaries

**Table 2 Top ten proteins detected by mass spectrometry from FKRP pull down.**

| No. | Unused | Total | Accession | Name |
|---|---|---|---|---|
| 1 | 160.45 | 160.45 | sp|Q8VDD5 | MYH9_MOUSE | Myosin-9 OS = Mus musculus GN = Myh9 PE = 1 SV = 4 |
| 2 | 60.11 | 60.11 | sp|Q9JMH9 | MY18A_MOUSE | Unconventional myosin-XVIIIa OS = Mus musculus GN = Myo18a PE = 1 SV = 2 |
| 3 | 21.6 | 46.44 | sp|Q61879 | MYH10_MOUSE | Myosin-10 OS = Mus musculus GN = Myh10 PE = 1 SV = 2 |
| 4 | 28.46 | 28.46 | sp|Q01853 | TERA_MOUSE | Transitional endoplasmic reticulum ATPase OS = Mus musculus GN = Vcp PE = 1 SV = 4 |
| 5 | 21.2 | 21.27 | sp|Q8CIG8 | ANM5_MOUSE | Protein arginine N-methyltransferase 5 OS = Mus musculus GN = Prmt5 PE = 1 SV = 3 |
| 6 | 11.52 | 11.55 | sp|Q60605 | MYL6_MOUSE | Myosin light polypeptide 6 OS = Mus musculus GN = Myl6 PE = 1 SV = 3 |
| 7 | 22.71 | 22.71 | sp|P05213 | TBA1B_MOUSE | Tubulin alpha-1B chain OS = Mus musculus GN = Tuba1b PE = 1 SV = 2 |
| 8 | 2.11 | 20.92 | sp|P68369 | TBA1A_MOUSE | Tubulin alpha-1A chain OS = Mus musculus GN = Tuba1a PE = 1 SV = 1 |
| 9 | 16.35 | 16.46 | sp|P56480 | ATPB_MOUSE | ATP synthase subunit beta, mitochondrial OS = Mus musculus GN = Atp5b PE = 1 SV = 2 |
| 10 | 6.84 | 6.86 | sp|Q8CGP2 | H2B1P_MOUSE | Histone H2B type 1-P OS = Mus musculus GN = Hist1h2bp PE = 1 SV = 3 |

**Table 3 Top ten proteins detected on mass spectroscopy from fibronectin pull down.**

| No. | Unused | Total | Accession | Name |
|---|---|---|---|---|
| 1 | 230.11 | 230.11 | sp|P35579 | MYH9_HUMAN | Myosin-9 OS = Homo sapiens GN = MYH9 PE = 1 SV = 4 |
| 2 | 49.63 | 49.63 | sp|P02751 | FINC_HUMAN | Fibronectin OS = Homo sapiens GN = FN1 PE = 1 SV = 4 |
| 3 | 32.51 | 32.51 | sp|P04264 | K2C1_HUMAN | Keratin, type II cytoskeletal 1 OS = Homo sapiens GN = KRT1 PE = 1 SV = 6 |
| 4 | 17.05 | 34.27 | sp|P35580 | MYH10_HUMAN | Myosin-10 OS = Homo sapiens GN = MYH10 PE = 1 SV = 3 |
| 5 | 16.15 | 16.33 | sp|P13645 | K1C10_HUMAN | Keratin, type I cytoskeletal 10 OS = Homo sapiens GN = KRT10 PE = 1 SV = 6 |
| 6 | 9.7 | 9.7 | sp|P35527 | K1C9_HUMAN | Keratin, type I cytoskeletal 9 OS = Homo sapiens GN = KRT9 PE = 1 SV = 3 |
| 7 | 6.2 | 6.2 | sp|P81605 | DCD_HUMAN | Dermcidin OS = Homo sapiens GN = DCD PE = 1 SV = 2 |
| 8 | 4 | 4.01 | sp|P04259 | K2C6B_HUMAN | Keratin, type II cytoskeletal 6B OS = Homo sapiens GN = KRT6B PE = 1 SV = 5 |
| 9 | 3.52 | 3.54 | sp|P60709 | ACTB_HUMAN | Actin, cytoplasmic 1 OS = Homo sapiens GN = ACTB PE = 1 SV = 1 |
| 10 | 2 | 2 | sp|P21333 | FLNA_HUMAN | Filamin-A OS = Homo sapiens GN = FLNA PE = 1 SV = 4 |

after fibronectin injection (Fig. 3f). This analysis revealed that collagen was expressed at the myoseptal boundaries of wild-type fish injected with sialylated or desialylated fibronectin. Furthermore, *fkrp*$^{-/-}$ fish injected with fully sialylated fibronectin possessed significantly more collagen at the MBM compared to uninjected mutants (Fig. 3f). By contrast, *fkrp*$^{-/-}$ mutant larvae injected with desialylated fibronectin failed to rescue the lack of collagen at the MBM evident in these mutants (Fig. 3f, g). The results confirmed that only sialylated fibronectin can rescue the collagen deposition deficits evident in *fkrp*$^{-/-}$ mutants. To test if these fibronectin rescue experiments could physiologically improve *fkrp*$^{-/-}$ mutant muscle function, passive force was measured. Injection of *fkrp*$^{-/-}$ larvae with natively glycosylated, but not desialylated, fibronectin rescued the ability of these mutants to transmit passive force when compared to sham-injected (PBS) *fkrp*$^{-/-}$ controls (Fig. 3h). This finding suggests that the fibronectin sialylation pathway could provide a potential novel target for therapy development for FKRP-deficient patients. Collectively, these results surprisingly reveal that the FKRP-dependent sialylation of fibronectin is necessary for fibronectin–collagen binding and maintenance of MBM integrity. Our data also have the potential to explain an unusual aspect of the FKRP clinical phenotype, namely the lack or late presentation in patients of joint contractures, stiffness of the joints that results from fibronectin–collagen accumulation, a pathology associated with the vast majority of other muscular dystrophies[1,25].

**Fibronectin Golgi localisation and sialylation requires Fkrp.** Next, we sought to determine the mechanistic basis by which FKRP regulates fibronectin sialylation. We examined three distinct possibilities. Firstly, we examined if FKRP could catalyse fibronectin sialylation directly. Although FKRP contains no protein domains that suggest that it could itself catalyse fibronectin sialylation, we specifically examined this possibility by desialylating fibronectin and providing it as a substrate for recombinant FKRP

protein and CMP-sialic acid donor substrate (Extended Data Fig. 9). Analyses of glycan addition by PGC–LC-MS/MS revealed that FKRP could not re-sialylate fibronectin, whereas a control using the recombinant sialyltransferase (ST6Gal1) could (Extended Data Fig. 9). Secondly, we investigated the hypothesis that a FKRP-dependent process could be required for sialyltransferase gene expression by undertaking comparative RNA Seq analyses in patient and control human myotubes (Extended Data Fig. 8). However, this analysis failed to detect any downregulation in the expression of any sialylation pathway gene. Intriguingly, ST6Gal1, one of the enzymes potentially responsible for the terminal α2,6-sialylation process that is defective in FKRP mutants, was found to be strongly upregulated in FKRP-deficient myotubes (Extended Data Fig. 8)[26]. This observation suggests that muscle cells may actively survey fibronectin sialylation, and regulate sialyltransferase levels accordingly. The final hypothesis we tested was that FKRP may act as an obligate binding partner for a yet-to-be-identified sialyltransferase or a sialyltranferase-binding protein. Such a model is supported by the observation that FKRP has previously been shown to exist in a large multimeric complex comprising several high-molecular-weight components within the Golgi[27]. Immunoprecipitation and mass spectrometry SWATH analyses detected no association between FKRP and known sialyltransferase enzymes (Table 2). However, it did reveal that non-muscle myosin10 bound to FKRP, an association considered significant, given that previous studies have revealed that myosin10 can be tethered through its carboxyterminus to the aminoterminus of glycosyltransferases including sialyltransferases[28,29]. Furthermore, reverse immunoprecipitation and mass spectrometry SWATH analyses assaying fibronectin from human control cells, revealed that myosin10 was also found associated with fibronectin (Table 3), a result confirmed by independent co-immunoprecipitation analyses using an anti-myosin10 antibody, which readily detects fibronectin in control and patient samples (Extended Data Fig. 9a). Furthermore, co-immunoprecipitation studies from the identical human patient

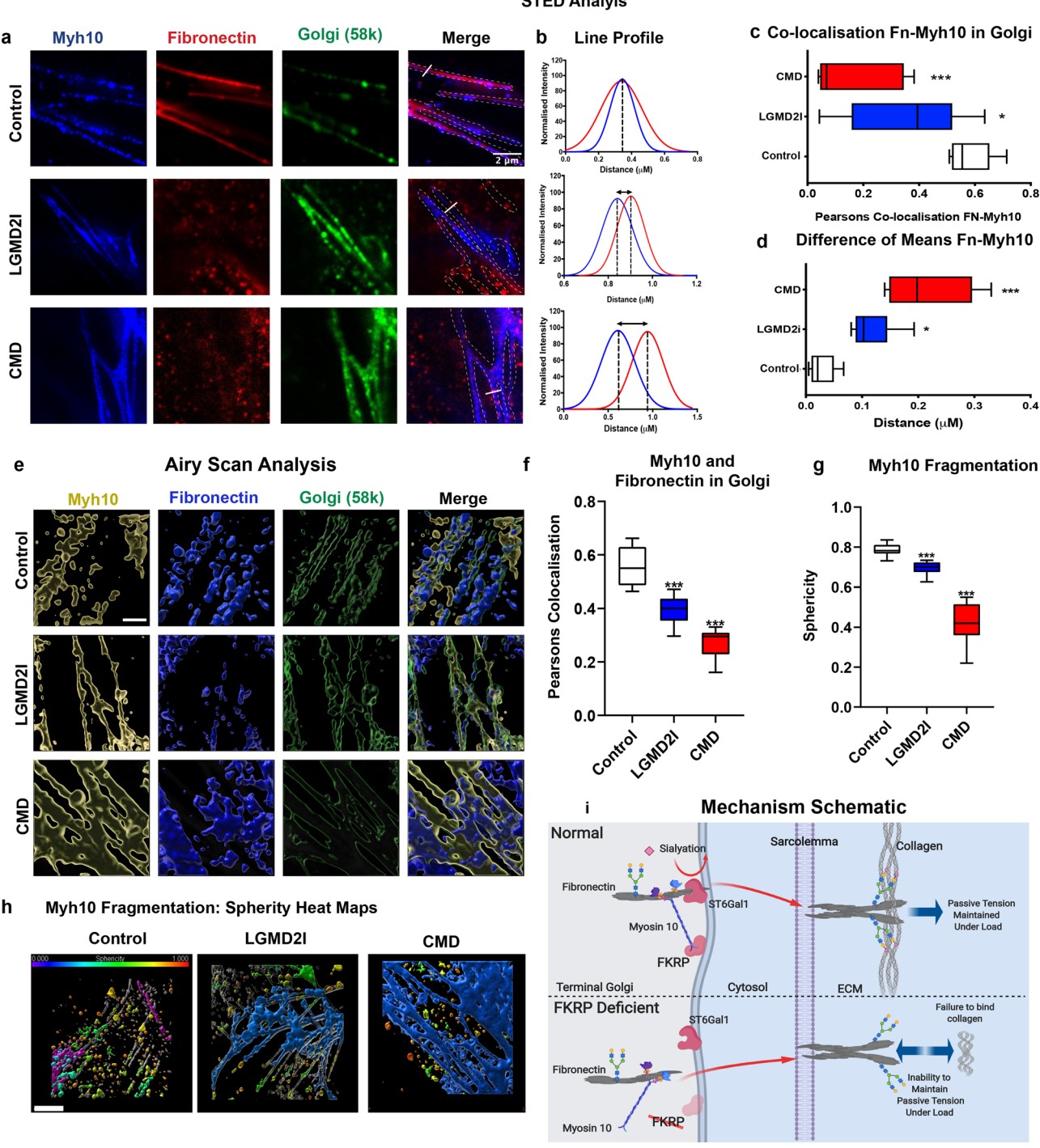

myotube membrane-enriched samples using a FKRP antibody readily detect Myosin-10 in control samples, but do not detect Myosin-10 in pulldowns conducted on patient cell lines (Extended Data Fig. 9b). Collectively, these results suggest a model whereby loss of FKRP function disrupts the localisation and tethering of fibronectin to Myosin-10 within the Golgi, a process critical for fibronectin maturation and sialylation, which in turn is required for collagen–fibronectin binding. In line with a possible role for myosin10 in fibronectin maturation, previous studies on myosin10 mutant mice have detected collagen deposition deficits in alveolar interstitial ECM[30–32]. Interestingly, the interstitial alveolar ECM in the lung is required for maintaining its elastic properties in a manner analogous to the MBM.

Since non-muscle myosins have also been shown to stabilise high-molecular-weight protein complexes, such as actin scaffolds in the trans-Golgi while they are being sialylated, we postulated that a similar mechanism may regulate localisation of fibronectin and its consequent sialylation in the Golgi[33,34]. Furthermore, Golgi dysfunction in dystroglycanopathy was hypothesised as a contributing factor for the heterogeneity of these disorders in Fukutin-deficient mouse cardiac cells[35]. To examine this question, confocal and stimulated depletion emission (STED) microscopy were used to determine if fibronectin was mislocalised in the trans-Golgi of patient cells (Fig. 4a and Extended Data Fig. 10a–d). Using this approach, fibronectin was found to be co-localised with myosin10 in the Golgi of control cells, a

**Fig. 4 The relative localisation of fibronectin and myosin10 is altered in the Golgi of FKRP-deficient cells.** Fibronectin and myosin10 protein localisation within the trans-Golgi, as determined by STED microscopy and Gaussian distribution line profile analyses, images representative of a minimum of three repeats separated by a minimum of a week and three technical repeats. **a** STED images of myosin10: blue, fibronectin: red and Golgi 58 k protein: green. Merged image marked for Golgi with dashed lines and white solid lines marking the location of the line profiles quantitated in **b**. **b** Individual line profiles, normalised to maximum and using Gaussian distribution curve in Fiji image analysis software, fibronectin red, myosin10 blue, means marked with dotted lines. **c** Co-localisation of myosin10 and fibronectin from the Golgi identified by a 58 K antibody stain, Pearson correlation analysis (*$P < 0.05$), (***$P < 0.0001$) ($n = 9$), box-and-whisker plot, middle line = mean, box = 95% confidence interval, error bars = SEM, one-way ANOVA analysis. **d** Quantitation of the difference between individual line profile means, analysed from Gaussian fit curves in **b** ($n = 9$), (*$P < 0.05$), (***$P < 0.0001$). Box-and-whisker plot, middle line = mean, box = 95% confidence interval, error bars = SEM. **e–h** Analysis of Airyscan data in Imaris image analysis software. **e** Images from the Airyscan rendered in 3D, myosin10 in yellow, fibronectin in blue and Golgi: Golgi Reassembly Stacking Protein 2 (GORASP2) in green, scale bar = 2 µm. **f** Pearson's correlation of fibronectin and myosin10 using Golgi marker: GORASP2 was used as a mask for analysis. Box-and-whisker plots, error bars represent a 95% confidence interval and middle box line represents mean (***$P < 0.0001$). One-way ANOVA analysis ($n = 9$ for each sample), one-way ANOVA analysis. **g** Sphericity of fibronectin within Golgi, Imaris generated surfaces and software analysis. Box-and-whisker plots, error bars represent a 95% confidence interval and middle box line represents mean (***$P < 0.0001$). One-way ANOVA analysis ($n = 9$ for each sample). **h** Airyscan data with cells rendered in 3D and statistically coded for sphericity in a heat map (with a value of one (red) indicating a perfect sphere) reveal the altered Golgi structure in patient cells, scale bar 2 µm. **i** Schematic of FKRP action. The schematic illustrates that in healthy control cells, FKRP is required to correctly localise myosin10 within the terminal Golgi, a process important for sialylation of fibronectin, that in turn regulates its binding to collagen. In FKRP deficiency, myosin10 is no longer anchored correctly and fibronectin is consequently not sialylated correctly. This lack of sialylation results in a failure of collagen–fibronectin binding at the MBM, which ultimately leads to a loss of MBM stability and an inability of individual muscle fibres to resist passive force.

distribution that was significantly altered in patient-derived cell lines (Fig. 4c), the severity of which correlated with the clinical phenotype of the patients from which cells were derived. Specifically, myosin10 was found to be less fragmented within patient cells compared to controls, when its localisation in the Golgi was rendered in 3D, suggesting that a loss of myosin10-associated functions occurs specifically within patient cells (Supplementary Video 1 and Extended Data Fig. 10c, d). Furthermore, Gaussian fit analyses of the STED-derived data revealed that fibronectin and myosin10 were mislocalised specifically within the trans-Golgi, which is the site within the Golgi where sialylation occurs (Fig. 4b, d). In addition, analysis of 3D-rendered Airyscan super-resolution data also confirmed that the relative localisation of fibronectin and Myosin-10 was altered within the trans-Golgi of patient cells when compared to controls (Fig. 4e–h, Extended Data Fig. 10c, d, and Supplementary Video 2). We conclude that FKRP is required to correctly localise fibronectin within the trans-Golgi, a process that is essential for its correct sialylation (Fig. 4e).

## Discussion

Overall, the results we describe here, taken together with previously published results[4], reveal that FKRP functions to regulate the two most prominent ECM cell-binding adhesion complexes in skeletal muscle[1,36]. Firstly, FKRP catalyses the addition of ribitol-phosphate, a noncanonical glycan in vertebrates, to α-DG, and thereby regulates a critical point in α-DG glycan maturation required for its binding to laminin[4]. Secondly, our results reveal that FKRP regulates fibronectin sialylation within the Golgi. In this case, we speculate that the negative charges imparted via sialylation are critical for the adherence of fibronectin to collagen via its gelatin-binding region, a positively charged $Zn^{2+}$ ion-binding domain[37], previously found to be important for collagen binding of other negatively charged macromolecules[38]. The interdependency of these two important MBM maturation processes on FKRP function may provide the mechanism via which muscle fibre stability at MBMs is matched to the passive tension-protective properties of the myofibres themselves[1,9,39]. More broadly, these findings highlight that the molecular basis of dystroglycanopathies extends beyond the DGC axis and provides a potential explanation for the broad clinical spectrum evident in this group of disorders. Furthermore, given the importance of fibronectin–collagen function in the maintenance of general

tissue integrity and in the progression of disease processes as diverse as cancer and fibrosis, these results have the potential to impact a diversity of pathological processes.

## Methods

**Zebrafish strains, maintenance and husbandry.** The strains used in the experiments were Tubingen (Tu) (referred to in the text as wild-type), $dag1^{hu3072}$ (referred to in the text as $dag1^{-/-}$)[40] and $fkrp^{pc36(+5bp)}$ (referred to in the text as $fkrp^{-/-}$, and is the allele used for the majority of experiments), $fkrp^{pc37(-exon3)}$, Tg (acta1:lifeact-GFP;mCherryCAAX). Adult zebrafish were kept as previously described[41] and bred under animal ethics breeding colony license number ERM:14481, issued by Monash Animal Services Ethics Committee. Embryos were maintained in E3 medium (5 mM NaCl, 0.2 mM KCl, 0.4 mM $CaCl_2$, 0.9 mM $MgCl_2$ and 2% methylene blue) in $ddH_2O$ and incubated at 28.5 °C. Embryos/larvae were staged as per described criteria[15] and were culled using 0.04% tricaine diluted in distilled water[40]. All experimental procedures were reviewed and approved by Monash Animal Services Ethics Committee prior to commencement under project ID: 5652.

**Generating *fkrp* mutant lines.** $fkrp^{pc36(+6pp)}$: mRNAs encoding a pair of Zinc-finger nucleases targeted to the *fkrp* locus (CompoZr™ Knockout Zinc Finger Nuclease, Sigma-Aldrich, lot number 03261240MN) were injected into single-celled zebrafish embryos at a final concentration of 50 ng/µl. To identify individual fish that were carrying *fkrp* mutations, injected fish were raised to sexual maturity and crossed to wild-type, DNA extracted from the resulting individual embryos using the REDExtract-N-Amp Tissue PCR Kit (Sigma-Aldrich, Catalogue no. XNATR-1KT) and screened by High-Resolution Melt analysis (HRM) on a Roche LightCycler® 480 II using LightCycler® 480 High-Resolution Melting Master mix (Catalogue No. 04909631001), with the following primers: fkrp melt-L TCATCTTCGGGTCAATCTGC and fkrp melt-R TCTCCCTGCAGAGCTGTG. Mutations were sequence-confirmed by BigDye sequencing at the Garvan Molecular Genetics Facility (Darlinghurst, NSW). Zebrafish were genotyped by sequencing using the following genotyping primers: fkrp_Forward: CGAGTTCA ACGTGCAGACAG and fkrp_Reverse: AGCTAAACAAAGGCCGATGA. $fkrp^{pc37(-exon3)}$: Alt-R CRISPR–Cas9 crRNAs were targeted to the start and end of exon 3 of the *fkrp* locus in zebrafish: fkrp1: TTGGCTCTTCCAGAGGGAGC, fkrp 2: GGAACTCAAATTTGGAGAAG (IDT) and injected into single-celled zebrafish embryos at a final concentration of 36 ng/µl together with Alt-R S.p. Cas9 Nuclease (IDT) at 0.5 µg/µL in 20 mM HEPES, 150 mM KCl, pH 7.5. To identify individual fish that were passing on *fkrp* mutations, injected fish were raised to sexual maturity and crossed to wild-type, DNA was extracted and genotyped using the following primers: Forward: fkrp_crispr_forward: TTTGCCAGGCTCTGTTAA and fkrp_cispr_reverse: CGACTTCTGTCTAGTCTT.

**qPCR.** RNA was extracted from larvae or cells to generate RNA for cDNA production as previously described[15]. In total, 20 ng of cDNA samples were run in triplicate with LightCycler® 480 SYBR Green I Master reaction mix (Roche) using the primers listed in Supplementary Table 1. Reactions were run on the Roche LightCycler® 480 Real-Time PCR Machine. CT values for each sample set were normalised against the housekeeper genes, before comparing the data sets.

**RNA sequencing and analysis.** In total, 600,000 cells from human myoblast cell lines CMD, LGMD2I and control, were harvested and RNA-extracted using RNAeasy Micro kit (QIAGEN, Cat 74004). RNA concentrations were measured using Quibit RNA HS Assay Kit (Life Technologies, Cat Q32855) on a Denovix Fluorometer. Samples were processed with 200 ng of the total RNA. PolyA RNA seq libraries were generated using Illumina TruSeq Stranded mRNA Sample prep, protocol 15031047 RevE Oct 2013. Libraries were prepared according to the Illumina standard RNA protocol. Each library was single-end with a 50-nt read length. Sequencing was performed on the Illumina HiSeq 3000. RNA library preparation and sequencing were carried out by the Monash University Medical Genomics Facility. Sequencing reads were filtered and trimmed with Trimmomatic (v 0.36, Phred score of six consecutive bases below 15, minimum read length of 36 nt)[42] and mapped to the human genome (GENCODE's GRCh38 primary assembly, annotation v24) with STAR (v 2.5.2b)[43]. Read-to-gene assignment was done with feature Counts (1) (v1.5.2, reverse-stranded). Only genes with more than five sequencing reads and one count per million of mapped reads in at least two samples were considered for further analysis. $Log_2$ counts per million of mapped reads were calculated on TMM-normalised[44] sample libraries using edgeR's[45] cpm function (prior.count = 0.5). Differential gene expression was performed with Degust webtool (http://degust.erc.monash.edu/), using limma-voom. Genes with a false discovery rate (FDR) < 0.05 were considered to be differentially expressed.

**Whole-mount in situ hybridisation (WISH).** Embryos were collected at 24 hpf, manually dechorionated, washed in PBS twice and then fixed for 2 h at room temperature in 4% PFA. Embryos were stepped through a methanol/PBS series and stored at −20 °C, until WISH was performed as previously described[46] using probes for *fkrp*[47]. Embryos were deyolked manually and heads removed for genotyping and bodies mounted in glycerol and imaged on an Olympus dotslide BX51 microscope. Whole embryos were imaged on a Zeiss stereo Discovery V.20 and AxioHr colour camera.

**Western blot and co-immunoprecipitation.** Larval heads were removed and used for genotyping, bodies were snap-frozen, homogenised in immunoprecipitation buffer (Pierce). In all, 30 µg of protein was loaded on a 4–12% Bis-Tris Protein Gel MES Buffer (ThermoFisher Scientific). Primary antibodies were prepared in TBS blocking buffer (LI-COR): fkrp: rabbit polyclonal 1:500 fkrp[48], 1:500, Beta-Tubulin-Loading Control (Abcam: ab6046), Golgi 50 kda, IgG rabbit (Abcam: ab6046) 1:10,000, incubated in secondary antibody IRDye 800CW goat anti-rabbit (LI-COR) 1:5000. Blots were scanned on a infrared scanner (Odyssey). Co-immunoprecipitation utilised 10ug of antibody, fibronectin or FKRP, and a protein-G dynabead kit (Thermofisher).

**Immunohistochemistry.** For retina-based experiments, humanely killed 5-dpf embryos were incubated in 4% PFA at room temperature for 3 h. The fish were incubated in 30% sucrose in PBS at 4 °C overnight. Fish were aligned in OCT-embedding medium and snap-frozen. Using a cryostat, samples were trimmed to the level of the eye by cutting in a transverse plain. Sections 12-µm thick were mounted on glass slides. Samples were incubated in primary antibodies: 43DAG (anti-β-DG mouse IgG 1:50, Novocastra), IIH6 (anti-α-DG mouse IgM, Merck-Millipore 05-593 1:50), pan-laminin (anti-rabbit IgG polyclonal 1:100, Sigma), fibronection (rabbit 1:200, Sigma) and Collagen-1 (rabbit, Abcam, 1:100) in PBST overnight at room temperature. Sections were incubated in secondary antibodies: (goat anti-mouse IgM Alexa Fluor 594, Invitrogen; 1:500, goat anti-mouse IgG Alexa Fluor 594, Invitrogen; 1:500), (donkey anti-rabbit IgG Fluor 594, Invitrogen; 1:500), DAPI (1 mg/ml, 1:10,000) for 20 min at room temperature and mounted with Mowiol (Sigma).

For whole-mount staining, fish were stained as previously described[41]. Fish were raised to 1–7 dpf in E3 medium with 0.003% 1-phenyl 2-thiourea (PTU) to clear pigment added at 22 hpf[49]. Primary antibodies were used as described above. Fish were mounted in ProLong® Diamond Anti Fade (Life Technologies).

Human myoblast lines were seeded either onto coverslips or eight-well STARSTED (Nümbrecht, Germany) chambers and fixed after 24 h using 4% PFA, then immunolabeled. Primary antibodies were fibronectin IgG1 rabbit (Sigma, F3648: 1:200), Golgi 58 k Goat polyclonal ab23932 (Abcam: 1:200), Golgi GORASP2 ab204335 (Abcam 1:200). Anti-non-muscle myosin IIB/MYH10 antibody [3H2] ab684, and the secondary antibodies used for confocal imaging as described above. For super-resolution experiments: Alexa488 (Invitrogen: 1:250), goat anti-mouse IgG Abberior580 1:250 and goat anti-rabbit IgG Abberior635 1:250 were used. Cells on coverslips were mounted in ProLong® Diamond Anti Fade (Invitrogen) and imaged.

**Confocal and super-resolution microscopy.** Whole-mount fish were imaged on Nikon C1 upright confocal with a ×20 objective. Twenty optical slices were imaged for each fish through the myotome centred on the anal pore between the skin and the notochord. A Z-projection was analysed for maximum fluorescence intensity on Fiji image analysis software; five measurements on five vertical myosepta were collected across a minimum of three technical and three biological replicates. Cells

were imaged in the same manner with the exception of a 40x objective and only a single slice was imaged.

Stimulated emission depletion microscopy (STED, Abberior Instruments GmbH, Göttingen, Germany) was carried out on a microscope equipped with an Olympus ×100 oil objective (UPlanSApo NA = 1.4) with a 1-watt 775-nm pulsed STED laser. Image acquisition for the Alexa647 fluorophore used STED laser power of 8% and 640-nm excitation laser power of 25%, and for the Alexa568 and 594 fluorophores, STED laser power of 33% and 561-nm excitation laser power of 30%, and the Alexa488 flurophore was imaged using the 488-nm excitation laser alone at 10%. Pixels were set at 30 nm for x and y for all channels. Line profile analyses were performed on images collected across three replicates.

A Zeiss LSM 980 Airyscan 2 with a ×63 oil C PlanApo 1.4NA objective was used to generate a 4 × 4 tiled area with a z-volume using the Airyscan 2 Multiplex SR-8Y mode, 3D Airyscan post-processing was performed followed by analysis in Imaris.

**Image analysis.** For STED-based experiments, line profiles were drawn through Golgi-imaged channel at random, and the fibronectin Myh10 intensity values obtained were normalised to the highest value before creating a Gaussian profile in Graphpad Prism 7. Co-localisation analysis was performed on the STED and confocal images using the Imaris (Bitplane) Coloc module. Golgi co-localisation to fibronectin ($n = 5$) from confocal images was investigated. Co-localisation analysis on the STED images were examined for the co-localisation of myh10 and fibronectin within the Golgi ($n = 9$).

**Transmission electron microscopy.** Larvae were fixed in Karnosky fixative (4% paraformaldehyde, 2.5% glutaraldehyde, 0.1 M Na-cacodylate, 5 mM $CaCl_2$, 10 mM $MgCl_2$ and distilled water)[50]. This was followed by post-fixation in osmium tetroxide plus potassium ferricyanide (1% OsO4, 1.5% K3Fe(III)(CN)6 and 0.065 M Na-cacodylate buffer). Larvae were dehydrated through ascending concentrations of acetone (50%, 70%, 90% and 100%). Embedding consisted of a solution of ascending concentrations of Epon-Araldite resin: propylene oxide, and the resin was polymerised at 60 °C for 48 h. Sections were cut and then stained with 2% uranyl acetate followed by 1% lead citrate. Images were taken on a Hitachi H7500 with a Gatan Multiscan 791 digital camera.

**Physiology and functional studies.** In total, 6 dpf larvae were anaesthetised in tricaine and placed in MOPS-buffered physiological solution. Small aluminium foil sheets were folded into clips on both sides of the trunk musculature of each fish. Holes were then punched in these aluminium clips, enabling the fish to be mounted horizontally onto an in vitro muscle physiology tester apparatus (model 1500 A, force transducer 403 A, Aurora Scientific, Ontario, Canada). One hook was attached to a force transducer and the other hook was anchored to a motor that allowed graduated fine-length adjustments. The optimum muscle length (Lo) was determined for maximum active force generation by applying single-twitch stimulations while lengthening step-wise, until the force detected no longer increased with stretch. The fish was stretched in increments of 5% of Lo and held at that length for 300 ms to measure the passive force response. The initial peak of the force-time graph was used for analysis. This was repeated up to 25% of Lo stretch for each zebrafish larva[51]. All measurements were carried out at room temperature.

**Culture of human myoblasts.** Three human myoblast lines were obtained from the European Biobank in Milan (Human Ethics ID: 5652): a healthy control and two muscular dystrophy lines: a male 15-month-old CMD patient with *FKRP* mutations: c.693 G > C, p.Ile478Thr and a male 4.5 years old with LGMD2I with *FKRP* mutations: c.826 C > A, p.Leu276Ile. Use of patient cell lines at The Australian Regenerative Medicine Institute, Monash University, was conducted with ethical oversight and review via the Monash University Human Research Ethics Committee project number CF14/3369–2014001793. Cells were cultured and differentiated to myotubes and allowed to mature for 5 days as per standard operating procedures outlined by the Euro Biobank: http://www.eurobiobank.org/biobanking-sops/ [accessed: January 3, 2018]. Cell lysates were collected with 0.25% Trypsin EDTA and spun down at 200 g for 5 min with myoblast media to neutralise the trypsin. Cell pellets were washed three times with 4 °C PBS. For imaging work and RNA-seq analysis, cells were cultured to 75% confluent myoblasts. Cells were differentiated to myotubes for 6 days for all mass spectrometry work, including glycan analysis and protein identification.

**Quantitative proteome analysis.** Larvae or cell lysates were homogenised in immunoprecipitation buffer (Pierce) and incubated with 10 µg of human fibronectin antibody (Sigma) bound to IgG beads as per the manufacturer's instructions. After the prescribed PBS and PBST wash steps, the pulled-down fraction was subjected to on-bead trypsin digestion. Briefly, beads were reconstituted to 100 µL in 100 mM triethylammonium bicarbonate containing 1% sodium deoxylcholate, reduced, alkylated and trypsin- digested. Digests were then separated from beads and purified by self-packed C18 tips. Samples were injected onto a C18 reversed-phase (RP) trap chip (ChromeXP, 120 Å, 3 µm) using a nanoLC 400 with cHiPLC

system (SCIEX) and separated on a 15-cm × 200-μm analytical RP chip column within 1-h acetonitrile gradients (2–35%). Eluting peptides were analysed using a TripleTOF 6600 (SCIEX) mass spectrometer in positive ion mode with a spray voltage of 2.5 kV. For data-dependent acquisition (DDA), a 0.25-s MS scan was followed by $20 \times 0.1$-s MS/MS scans of the most intense ions in the MS scan. For SWATH acquisition, 50 fixed 10 m/z windows (800–1300 m/z) were used. A 0.1-s MS scan was followed by $50 \times 0.06$-s SWATH MS/MS acquisitions at the range between 100 and 2000 m/z. DDA data were processed and searched through a Mascot searching engine. The result was used to generate a spectral library for SWATH quantitative analysis. SWATH data were analysed using Skyline. Integrated peak area of the top five most abundant transitions were summed as the quantitative value of each protein-specific peptide. The values of collagen-specific peptide (FTYTVLEDGCTK) were normalised to the fibronectin-specific peptide (YEVSVYALK) to obtain the relative abundance values.

**Glycan release and MS analysis.** Fibronectin was pulled down as described above and fractions separated in SDS-PAGE and electroblotted to the PVDF membrane. After staining, fibronectin bands were isolated for glycomic analysis according to established protocols[23]. Membrane pieces were washed by water and blocked by 1% polyvinylpyrrolidone (PVP40, Sigma). In total, 2.5 U of PNGase F (Roche) was added to each sample, and the reaction was conducted at 37 °C overnight. N-glycans were harvested from solution and were reduced by 1 M sodium borohydride in 50 mM potassium hydroxide at 50 °C for 3 h. O-glycans were then released from membranes by 0.5 M sodium borohydride in 50 mM potassium hydroxide at 50 °C for 16 h. The samples were desalted by Dowex X8 cation exchange resin and purification was carried out using a hand-packed porous graphitised carbon tip. Glycomic profiling of released glycans was conducted by PGC–LC–MS/MS as previously described[23].

**Collagen–fibronectin SPR.** Biacore analyses were carried on a T200 (GE Healthcare) using the dedicated control and evaluation software. Human fibronectin (Millipore EMD) and human collagen-1a (Sigma) were immobilised onto individual flow cells onto the surface of a CM5 Chip (GE Healthcare) by 1-ethyl-3-(3-dimethylaminopropyl) carbodiimide/N-hydroxysuccinimide (EDC/NHS) coupling at 50 μg/ml in NaOAc buffer, pH 4.0 (collagen) or pH 4.2 (fibronectin). Unreacted sites were capped with 1 M ethanolamine. Control surfaces were prepared in parallel by activation (EDC/NHS) and deactivation (ethanolamine) of the flow cell. Serial (doubling) dilutions of analyte from 100 to 1.57 nM were prepared in Running Buffer (10 mM HEPES, pH 7.4, 150 mM NaCl and 0.005 (w/v)% Tween 20) and injected over the control and cognate ligand surface for 2 min at 30 μl/min. The dissociation was monitored for 10 min and double regeneration carried out with a 30-s pulse of 0.1 M glycine-HCl, pH 2.0, followed by a 30-s pulse of 25 mM NaOH/0.5 M NaCl. For each experiment, seven concentrations, one in duplicate, and a zero (buffer only), were performed. Sensorgrams were double-reference-subtracted from the control flow cell and a buffer-only injection. Desialylation was performed 'on-chip' by flowing 6 U/μl neuraminidase (NEB) in 1× GlycoBuffer1 over the fibronectin surface (1 μl/min) for 4.87 h and continual incubation on the surface overnight (37 °C), after which multiple pulses (30 s) of 25 mM NaOH/0.5 M NaCl (30 μl/min) were injected to remove any associated neuraminidase. The following treatment dilutions of collagen were prepared in a manner similar to that previously described[52].

**Sialyltransferase assays.** Coding sequence of full-length human FKRP with C-terminal FLAG tag was synthesised (IDTDNA) and ligated into pGen2.1 vector to generate pGen2.1-hFKRP. In all, 320 μg of pGen2.1-hFKRP was used to transfect 160 mL HEK293T cells at a density of $2.5 \times 10^6$ cells/mL. Ninety-six hours post transfection, cells were pelleted and solubilised in 20 mL 25 mM HEPES, pH = 7.5, 150 mL KCl, 1× protease cocktail and 1% Triton X-100. Lysate was then clarified by centrifugation at 5000×g for 10 min. The supernatant was loaded on the anti-FLAG M2 agarose gel (Sigma). After washing by 5× column volume of PBS containing 500 mM NaCl, bound hFKRP was eluted by 0.1 M glycine-HCl and the pH was immediately brought back to 7.5 by 0.5 M HEPES, pH 7.5. Eluted hFKPR was concentrated and buffer-exchanged to 25 mM HEPES, pH 7.5, and 50 mM KCl. MS-based sialyltransferase assays were conducted in a final volume of 100 μl containing 25 mM MES, pH 6.4, 20 mM MgCl₂, 2 mM CaCl₂, 5 μg of desialylated fibronectin using neuraminidase (NEB) as per the manufacturer's instructions, 1 mM CMP-Neu5Ac, 5 μg of desialylated fibronectin and the enzyme to be tested. For negative controls, water was used instead of CMP-Neu5Ac. Reactions were conducted at 37 °C for 6 h. Reactions were brought to 1% sodium deoxycholate and 50 mM triethylammonium bicarbonate, reduced by 2.5 mM DTT and alkylated by 5 mM IAA. In total, 0.5 μg of sequencing-grade trypsin was added and incubated at 37 °C for 16 h. Tryptic digests were brought to 1% formic acid to precipitate sodium deoxycholate and spun at $13,000 \times g$ for 10 min. Supernatants were transferred to new tubes and dried in a Speedvac (Thermo Fisher Scientific). Digested peptides were resuspended in 2% ACN containing 0.1% formic acid and injected for LC–MS analysis. A SWATH method was used for data acquisition and analysis. Sialyltransferase activity was monitored by quantitative analysis of sialy-lation on a fibronectin glycopeptide (LDAPTNLQFV**N**ETDSTVLVR). The major glycoform on this glycopeptide from desialylated fibronectin is a non-sialylated

biantennary complex-type N-glycan and provides up to two sites for sialylation[53]. Changes of relative abundance among three species, namely, the glycopeptide-carrying biantennary complex N-glycans with no sialic acid (S0), one sialic acid (S1) and two sialic acids (S2), were analysed by our SWATH workflow.

**Fibronectin rescue and vasculature analysis.** Fibronectin protein 2 mg/ml (Millipore EMD) was either untreated or treated by desialylation using neuraminidase (NEB) as per the manufacturer's instructions. Fibronectin protein 1 mg/ml was tagged using Alexa Fluor™ 488 Microscale Protein Labelling Kit (ThermoFisher Scientific) as per the manufacturer's instructions. Tagged fibronectin was then diluted in 300 mM KCl and phenol red dye injection indicator in a 1:7:2 ratio. In total, 5 nl of the 100 μg/ml solution was then injected into the yolk of up to 1 hpf embryos. Fish were raised to 5 dpf in E3 medium with 0.003% 1-phenyl-2-thiourea (PTU) added at 22 hpf to clear pigment. Fish were raised in the dark to prevent bleaching of the fluorescently labelled protein. Fish were stained and imaged as described above. A cohort of the rescued fish were kept for passive force physiology analyses as described above. Experiments were blinded for each analysis and fish individually genotyped post experimentation. To interrogate vasculature, a working injection mix (0.1% EBD or 1% fluorescent dextran) was diluted using Ringer's solution prior to injection of 5 nl into the pericardial vein of 5 dpf larvae pre-treated with PTU at 22 hpf. After injection, zebrafish embryos were placed in Ringer's solution and incubated in the dark at 28 °C for 6 h before live mounting for imaging.

**Reporting summary.** Further information on research design is available in the Nature Research Reporting Summary linked to this article.

## Data availability
The RNA-seq dataset has been deposited at the NCBI Gene Expression Omnibus (GEO) repository under accession number GSE163213. The raw data from PGC–LC-MS/MS N-glycan analysis of fibronectin isolated from control and patient myoblast cell lines (Fig. 3) and whole-cell lysates (Supplementary Fig. 6) were submitted to GlycoPOST[54,55] under the ID GPST000159. The data are accessible at the following URL: https://glycopost.glycosmos.org/preview/15008940036038edf6af4e9 Pin code PIN 4292. Human cell lines can be obtained through the European Biobank http://www.eurobiobank.org. Other data and materials are available from the corresponding author (PDC) upon reasonable request. Source data are provided with this paper.

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

## Acknowledgements

This work was supported by the National Health and Medical Research Council of Australia (NHMRC) (PDC, NP: APP1127741, PDC: APP1136567, APP3151883). Australian Research Council Linkage (PDC: LP120100281) in collaboration with Sigma-Aldrich and LIEF Grant (LE150100110). The Australian Regenerative Medicine Institute is supported by grants from the State Government of Victoria and the Australian Government. We acknowledge J Callaghan, I Harper, Monash Micro Imaging Facility, J Clark and A Costin, Monash Electron Microscopy Facility, L Kautto, Australian Proteomic Analysis Facility, W Moore, J Hilton and J Manneken, Aquacore core staff, Australian Regenerative Medicine Institute and Monash University Micromon sequencing facility for technical support. We also thank G Lynch and M Anderson for paper critiques.

## Author contributions

A.J.W.: contributed to all areas of writing the paper, analysed data, prepared figures and designed and contributed to all experiments. C.H.L.: contributed to all areas of biochemistry in the paper, including analysed the data, prepared figures and designed and contributed to all biochemistry experiments. M.L.: responsible for physiology experiments and analysis. K.N.: contributed additional technical runs for glycan analysis, aided figure preparation and editing of the paper. S.A.: contributed experimental workup for RNA seq and transfection of C2C12 cells. F.R.: analysed RNA-seq data, prepared figures. A.S.: generated ZFN FKRP allele and genetically characterised line. L.M.: generated ZFN FKRP allele and genetically characterised line, contributed plasmids and cloning design advice. P.J. and S.D.: contributed retina experimental work and analysis, P.J. also was involved in all drafts of paper writing. C.S. and L.H.: contributed the data for all experiments involving zebrafish. C.K.: COIP mass spec experiments. P.C.: Biacore data and analysis. S.G. and M.M.: contributed patient cells and contributed to the design of experiments where used. A.F.: design and analysis of STED experiments. N.P.: contributed to the design of all biochemistry experiments and analysis. General guidance on all aspects of experimentation in the paper and writing of the paper. P.D.C.: contributed to the design and analysis of all experiments and writing of the paper.

## Competing interests

The authors declare no competing interests.
