## [Peer Review File · Nature Communications]

Reviewers' Comments:

Reviewer #1:

Remarks to the Author:

Overall Comments: The manuscript by Wood and colleagues focuses on the generation and characterization of a zebrafish model of FKRP-deficiency. FKRP pathogenic variants in human patients can lead to either LGMD2I or a more severe Walker Warburg Syndrome (WWS) resulting in severe muscle and brain pathologies. Currently, the only identified function of the FKRP protein is as a glycosyltransferase where it transfers ribitol-5-phosphate to the alpha-dystroglycan (αDAG1) protein. The authors performed a thorough and well-executed series of experiments that demonstrates for the first time the dependency of N-glycan sialylation of fibronectin on FKRP. Using an arsenal of high-resolution imagery and biochemical techniques, they show that loss of FKRP results in aberrant fibronectin function (due to reduced N-glycan sialylation) which results in loss of collagen at the MBM. Ultimately, this leads to loss of passive force. The authors further demonstrated that they could restore collagen and passive force by injecting sialylated fibronectin. Moreover, they investigated how FKRP could influence sialylation status of fibronectin by measuring direct and indirect measures. They identified a non-muscle myosin, myosin10 as a possible key player in this interaction, that requires FKRP in order to form a complex that properly sialylates fibronectin in the trans Golgi.

Overall, this is a well-written manuscript, except for a few gene nomenclature issues, with a key finding on the modification of fibronectin requiring functional FKRP protein which is disrupted in LGMD2I and WWS. However, there are key aspects of their findings that need clarification to fully understand the nature of the regulatory steps of FKRP on fibronectin sialylation.

Major Comments:

1. The authors state at that α-dystroglycan glycosylation status has been shown that it does not track with disease progression or severity. Does the N-glycan sialylation status of fibronectin track with either disease progression or severity?
2. In human samples (looking at LGMD2I and CMD patient cells), they found significant reduction in N-glycan sialylation compare to healthy controls in two peaks, that are composed of NeuAc1Hex5HexNAc4 and NeuAc2Hex5HexNAc4. Can the authors comment whether they believe both sites are important for the interaction with collagen? Or would partial sialylation of just one of these sites be enough to restore collagen interaction?
3. Increasing glycosylation status of α-dystroglycan has been shown to be beneficial in fkrp-deficient backgrounds. Can the authors discuss the relevant therapeutic targets that their study now opens up for patients with LGMD2I or CMD? Do they believe that somehow increasing sialylation of fibronectin can be a therapeutic target?
4. Were any neurological and/or vascular phenotypes observed in the fkrp mutant zebrafish? This has been reported in other reports and it would be good to validate the occurrence of these pathological findings. Bailey et al., *Skeletal Muscle*, 2019 reported altered NMJ formation in fkrp mutant zebrafish, the authors should comment on whether fibronectin disruption affects NMJ formation and/or myofiber detachment potentially independent of FKRP expression functional status.
5. Line 53 – Since the authors are talking about human gene, it should be FKRP not fkrp.
6. The correct zebrafish gene name for dystroglycan is dag1. The authors should be consistent in their naming and change “dag” to “dag1” in the manuscript.
7. RNA-sequencing (and ideally mass-spectrometry data) should be deposited in a public database

for free access.

Minor Comments:

1. Line 89 – "...loss of function15, we postulated..." Move the comma to after "function".
2. Line 129 – "FKRP-deficient" missing hyphen.
3. Line 219 – Remove the " `s" on fibronectin.
4. Line 734 – Typo; describer is supposed to be "described".
5. Figure 1 – the names and colors used to describe the groups are confusing and misleading. Use of the same colors to represent different things in the same figure is confusing to the reader. Also, consider using "wild type" or WT instead of fkrp+/+ to be consistent with in-text writing.
6. All figures – I am not a fan of the showing points outside the 95% confidence range but I will leave this up to the authors' discretion.
7. All figures – remove the two-sided arrow to indicate comparison between groups and use brackets instead – cleaner look.
8. The correct abbreviation is LGMD2I (with a capital I) or the newer abbreviation LMGDR9.

Reviewer #2:

Remarks to the Author:

Wood et al. reported that FKRP, one of causative genes of dystroglycanopathies, has a new function. They used FKRP-mutated zebrafish and human patient myoblasts, and they found FKRP-dependent sialylation, specific N-glycans, of fibronectin is important for the formation of fibronectin-collage axis. Based on the results, they proposed sialylation of N-glycan-fibronectin regulates muscle pathology in muscular dystrophy. However, to reach the conclusions, the authors should provide more convincing data.

- 1) The authors considered three possibilities of FKRP-dependent sialylation and thought Fig. 4e model is most possible. However, they did not show direct binding myosin10 and FKRP. Direct binding data of normal and mutant FKRP proteins to myosin10 is necessary. It is also necessary to show direct binding of fibronectin-myosin10 because they performed proteomic analysis of fibronectin-pull down fraction and found myosin10 as one of candidate proteins. It also requires to perform similar proteomic experiments using the Golgi-enriched fraction, or at least after removal of cytosol.
- 2) The authors claimed myosin10 localized in the luminal side of Golgi. However, it could not conclude by STED study. Actually, location of Golgi(58k) is reported in the cytosol side of Golgi. They should give concrete data of location myosin10 by different methods.
- 3) The authors concluded that FKRP did not catalyze fibronectin sialylation directly (Ex Fig.9). line 892, there is no description how to prepare enzymes? Does this FKRP protein have ribitol-P-transferase activity?
- 4) It is of interest that the authors found the increment of ST6Gal1 expression in FKRP mutants although they tried to find down regulated genes. However, it is unclear why they could not find any difference total sialylation (Ex Fig. 6) in control and patients although ST6Gal1 increased?

- 5) Lines 824, 825, pathological information including severity of CMD patient and LGMD2I patient was not described at all. They may be different. IIH6 staining patterns of both should be shown. However, sialylation of both fibronectin did not show any difference (Fig. 3b), suggesting that sialylation level of fibronectin does not correlate directly with disease severity. By the way, in Fig. 3a, Red box may be Blue box. Line 479, g should be removed. It is noted that this review could not see the increase of asialobianntenary N-glycan peak in patients. Why?
- 6) Line 240, CDP-ribitol should be ribitol-phosphate.
- 7) Lines 553~555, this explanation is wrong, because fukutin and FKRP sequentially contributed the a-dystroglycan-glycan formation.
- 8) Two reference sections could be found (page 12 and page 48). The latter one may be correct.
- 9) Fig. 4, why fibronectin staining patterns of patients are different from control?
- 10) Table 1, why the authors did not examine myosin 9? Refs 32 and 33 are related to myosin 9 (non-muscle myosin IIA).
- 11) Ex Fig.1, i, j and k are missing.
- 12) Ex Fig. 2c'', Laminin intensity data should be provided instead of FWHM.
- 13) Ex Fig. 7 legend is insufficient description. This suggests sialylation-independent binding of fibronectin-collagen also contributed?
- 14) First appearance of MBM (line 75). There is no explanation.
- 15) There are many type-errors and reference numbers (e.g. line 673, ref 439). These hampered the review process.

Manuscript: NCOMMS-20-10420 FKRP-dependent glycosylation of fibronectin regulates muscle pathology in muscular dystrophy.

Below we systematically address each of the reviewer's concerns. Corresponding text changes are highlighted in blue within the manuscript.

Reviewer 1:

Overall Comments: The manuscript by Wood and colleagues focuses on the generation and characterization of a zebrafish model of FKRP-deficiency. FKRP pathogenic variants in human patients can lead to either LGMD2I or a more severe Walker Warburg Syndrome (WWS) resulting in severe muscle and brain pathologies. Currently, the only identified function of the FKRP protein is as a glycosyltransferase where it transfers ribitol-5-phosphate to the alpha-dystroglycan (aDAG1) protein. The authors performed a thorough and well-executed series of experiments that demonstrates for the first time the dependency of N-glycan sialylation of fibronectin on FKRP. Using an arsenal of high-resolution imagery and biochemical techniques, they show that loss of FKRP results in aberrant fibronectin function (due to reduced N-glycan sialylation) which results in loss of collagen at the MBM. Ultimately, this leads to loss of passive force. The authors further demonstrated that they could restore collagen

and passive force by injecting sialylated fibronectin. Moreover, they investigated how FKRP could influence sialylation status of fibronectin by measuring direct and indirect measures. They identified a non-muscle myosin, myosin10 as a possible key player in this interaction, that requires FKRP in order to form a complex that properly sialylates fibronectin in the trans Golgi.

Overall, this is a well-written manuscript, except for a few gene nomenclature issues, with a key finding on the modification of fibronectin requiring functional FKRP protein which is disrupted in LGMD2I and WWS. However, there are key aspects of their findings that need clarification to fully understand the nature of the regulatory steps of FKRP on fibronectin sialylation.

Major Comments:

Q1.1. *“The authors state at that α -dystroglycan glycosylation status has been shown that it does not track with disease progression or severity. Does the N-glycan sialylation status of fibronectin track with either disease progression or severity?”*

Response. Our original data did detect a dramatic drop, but of similar magnitude, in the sialylation levels of fibronectin in both of the patient samples tested (Figure 3a, b). However, although this data did not show a difference in sialylation levels between the two patient samples, we did observe a highly significant difference in fibronectin-collagen binding between these two samples, that correlated with clinical severity (Fig 3c). This was also the case for the Golgi defects we characterised (Fig 4), both for the Myosin10-fibronectin co-localisation and the Myosin10 fragmentation phenotype, which tracked with clinical severity. Thus, although we could not resolve disease-specific severity differences in the levels of fibronectin sialylation, we believe it likely that there are patient specific changes in sialylation levels on fibronectin that we have not been able to detect with current approaches. Given that only 3 of the 7 potential biantennary glycan sites on fibronectin reside in the collagen binding domain, and that there is a limit to the amount of fibronectin we can pull down from patient myoblast samples, our N-glycan MS/MS technique may not have been sensitive enough to detect subtle changes in fibronectin sialylation that may occur on individual functionally important sites. In an attempt to resolve this question, we undertook glycopeptide analysis of

fibronectin derived from cultured patient cells with a view to improve our sensitivity of detection of sialylation levels on fibronectin glycosylation sites within the collagen binding domain. Although the Packer lab is one of the global leaders in undertaking such analyses we were unable to provide data that distinguishes between the two stages of the disease in terms of fibronectin sialylation. using this technically challenging approach. Unfortunately, this technology works well for high concentrations of protein obtained from traditional protein purification techniques, not rare muscle patient samples with limited passage potential. Nevertheless, the decrease seen in both patients, relative to the control, does implicate fibronectin sialylation as a significant factor in the disease. We have now also added additional text to the results at line 151-161 and line 233 page 10 explaining the cellular phenotypic tracking with clinical severity more clearly.

Q1.2. *In human samples (looking at LGMD2I and CMD patient cells), they found significant reduction in N-glycan sialylation compare to healthy controls in two peaks, that are composed of NeuAc1Hex5HexNAc4 and NeuAc2Hex5HexNAc4. Can the authors comment whether they believe both sites are important for the interaction with collagen? Or would partial sialylation of just one of these sites be enough to restore collagen interaction?*

Response. The reviewer raises an interesting question about the sialylation of the mono- and di-biantennary glycans released from the cellular proteins. However, the characterization of the glycans was carried out on *N*-glycans released from the proteins and as such we unfortunately have no information on the site occupancy of these two structures on the protein.

Q1.3. *Increasing glycosylation status of α -dystroglycan has been shown to be beneficial in *fkrp*-deficient backgrounds. Can the authors discuss the relevant therapeutic targets that their study now opens up for patients with LGMD2I or CMD? Do they believe that somehow increasing sialylation of fibronectin can be a therapeutic target?*

Response. This is an interesting point the reviewer makes and is the reason we carried out the rescue experiments by injecting *fkrp*^{-/-} mutant fish with non-sialylated and fully-sialylated fibronectin. From these analyses, presented in Fig. 3 of the original submission, it is clear that fully-sialylated fibronectin rescues the zebrafish *fkrp* mutant model, so indeed we think increasing sialylation of fibronectin can be a valid therapeutic approach. However, we know from previous work with laminin that these full-length proteins are unlikely to be used in the clinic as they are difficult to handle and deliver, despite providing excellent proof of principle in animal models. We rather envision investigating a possible therapeutic strategy that stabilises Myosin10 in the Golgi to improve fibronectin sialylation. We have now added a discussion of the therapeutic relevance of increasing fibronectin sialylation to page 8 line 180 of the revised manuscript in line with the reviewer's request.

Q1.4. *Were any neurological and/or vascular phenotypes observed in the *fkrp* mutant zebrafish? This has been reported in other reports and it would be good to validate the occurrence of these pathological findings. Bailey et al., *Skeletal Muscle*, 2019 reported altered NMJ formation in *fkrp* mutant zebrafish, the authors should comment on whether fibronectin disruption affects NMJ formation and/or myofiber detachment potentially independent of FKRP expression functional status.*

Response. The reviewer is indeed correct that a number of authors have reported vascular defects in zebrafish using a *fkrp*-targeting morpholino approach and this finding is also reported in Bailey et al. with additional interesting observations on NMJ formation. However, all these analyses were performed on *fkrp* morphants not on germline mutants which form the basis of our analyses. In line with the reviewer's comments we have further examined our *fkrp* germline mutants for vascular and neuromuscular defects. We used angiography with injected

Fluorescent-Dextran and independently evaluated the vasculature of Evans blue dye injected larvae to examine vascular permeability with the myotome. To study neuronal integrity, we stained for motoneuron arborescences and junctions in the myotome. None of these analyses showed vascular and neuromuscular disruption independent of the muscle loss or MTJ disruption within the myotome of *fkrp* mutants. The only time we could see defects in vascular or neuromuscular pattern were in regions of muscle loss or myosepta disruption in *fkrp* mutants. This new data is presented in Extended Data Figure 2f-g'' for the vascular analyses and in Extended Data Figure 3j, j' for neuronal defects. There are a number of potential explanations for this discrepancy such as the propensity for off target effects when morpholinos are used or the genetic compensation that is triggered by germline mutants that is not generated by morpholino knockdown¹. It is unclear at this point which explanation is the most likely. We have added text to manuscript explaining these new data points at pg4.

In terms of neurological defects, we also wish to point out that in the original submission we also carried out a detailed analysis of the retina which of course is part of the nervous system, given the muscle eye brain phenotype of some FKRP patients. In zebrafish, the retina replicates the neuronal layers in the human eye and as expected we did find abnormalities in basement membranes and modest defects in retinal layering in our mutants.

In regards to a potential role for fibronectin in NMJ formation, a search of the literature suggests no obvious mechanism for a role of fibronectin in NMJ formation. By contrast fibronectin is one of the primary components of fiber attachment in the myotome acting initially as a scaffold at the MTJ and later as a regulator of passive tension through control of myotome elasticity. Hence the loss of passive tension and MTJ boundary crossing phenotype we describe. However, it is highly likely that fibronectin's scaffolding role is multidimensional in this context, and almost certainly some of the roles it plays at the MTJ will be independent of its FKRP-mediated collagen binding activity.

1. El-Brolosy MA, Kontarakis Z, Rossi A, Kuenne C, Günther S, Fukuda N, Kikhi K, Boezio GLM, Takacs CM, Lai SL, Fukuda R, Gerri C, Giraldez AJ, Stainier DYR. Genetic compensation triggered by mutant mRNA degradation. *Nature*. 2019 Apr;568(7751):193-197.

Q1.5. *Line 53 – Since the authors are talking about human gene, it should be FKRP not fkrp.*

Response. Text altered as suggested.

Q1.6. *The correct zebrafish gene name for dystroglycan is dag1. The authors should be consistent in their naming and change “dag” to “dag1” in the manuscript.*

Response. Text altered throughout as suggested.

Q1.7. *RNA-sequencing (and ideally mass-spectrometry data) should be deposited in a public database for free access.*

Response. We are very happy to deposit all the data sets in public repositories. Specifically, the RNA-seq dataset has been deposited at the NCBI Gene Expression Omnibus (GEO) repository under accession number GSE163213 with a link that will be made public on acceptance of the manuscript. The raw data from PGC-LC-MS/MS N-glycan analysis of fibronectin isolated from control and patient myoblast cell lines (Figure 3) and whole cell lysates (Extended Figure 6) has been submitted to GlycoPOST under the ID GPST000159. The data is accessible at the following URL upon acceptance of the manuscript: <https://glycopost.glycosmos.org/preview/150345145fd6fcbaa8caa>. Statements outlining how these data sets can be accessed have been made within the methods on page 52.

Minor Comments:

Q1.9. Line 89 – “...loss of function15, we postulated...” Move the comma to after “function”.

Response. The text has been altered as requested.

Q1.10. Line 129 – “FKRP-deficient” missing hyphen.

Response. The text has been altered as requested.

Q1.11. Line 219 – Remove the “ ‘s” on fibronectin.

Response. The text has been altered as requested.

Q1.11. Line 734 – Typo; describer is supposed to be “described”.

Response. The text has been altered as requested.

Q1.12. Figure 1 – the names and colors used to describe the groups are confusing and misleading. Use of the same colors to represent different things in the same figure is confusing to the reader. Also, consider using “wild type” or WT instead of *fkrp*^{+/+} to be consistent with in-text writing.

Response: We agree with the reviewer the colors on the force graph (figure 1e) are confusing when compared with that of the boundary crossing and fibre detachment analysis (figure 1c and d) and we have altered figure 1e in line with the reviewer’s suggestion. In terms of the allele designation and the use of wildtype/WT, we have used the allele designation of *fkrp*^{+/+} to signify sibling rather than WT status. We use the designation wildtype/WT only when we refer to the Tübingen wildtype strain and *fkrp*^{+/+} when we are referring to a genotyped sibling control where heterozygotes have been excluded from analysis. The broad use of the wildtype designation could potentially mask the reader to the technical design of some of our experiments.

Q1.13. All figures – I am not a fan of the showing points outside the 95% confidence range but I will leave this up to the authors’ discretion.

Response. We understand the reviewer’s point here, but we believe it is Nature editorial policy to preferably represent all data points on graphs and have tried to adopt this policy in our graphical representations.

Q1.14. All figures – remove the two-sided arrow to indicate comparison between groups and use brackets instead – cleaner look.

Response. We removed all two-sided arrows except for the ones used on the passive force graphs (Fig 2g) where brackets may not convey the analysis.

Q1.15. The correct abbreviation is LGMD2I (with a capital I) or the newer abbreviation LMGDR9.

Response: The text has been altered as requested. We have stuck with the LGMD2I nomenclature as it is the most prevalent in the literature but are also happy to use the newer LMGDR9 designation if the reviewer believes it is more accurate to do so.

Reviewer 2

Wood et al. reported that FKRP, one of causative genes of dystroglycanopathies, has a new function. They used FKRP-mutated zebrafish and human patient myoblasts, and they found FKRP-dependent sialylation, specific N-glycans, of fibronectin is important for the formation

of fibronectin-collage axis. Based on the results, they proposed sialylation of N-glycan-fibronectin regulates muscle pathology in muscular dystrophy. However, to reach the conclusions, the authors should provide more convincing data.

Q2.1. *The authors considered three possibilities of FKRPs-dependent sialylation and thought Fig. 4e model is most possible. However, they did not show direct binding myosin10 and FKRPs. Direct binding data of normal and mutant FKRPs proteins to myosin10 is necessary. It is also necessary to show direct binding of fibronectin-myosin10 because they performed proteomic analysis of fibronectin-pull down fraction and found myosin10 as one of candidate proteins. It also requires to perform similar proteomic experiments using the Golgi-enriched fraction, or at least after removal of cytosol.*

Response. In line with the reviewers comments we have now included a co-immunoprecipitation experiment to demonstrate more directly the binding of Myosin10 to FKRPs. To accommodate the reviewer's request in regard to the use of a cytosol excluded fraction, we performed pull-down experiments on membrane enriched fractions that had the cytosol removed. Western blot analysis revealed that FKRPs co-immunoprecipitation of patient and control primary myotube membrane preparations readily detects Myosin10 in control samples but does not detect Myosin10 in FKRPs pull downs conducted on patient cell lines. By contrast, Myosin10 co-immunoprecipitation experiments, using the identical fractions, readily detected Fibronectin in both control and patient samples. Both results are in line with the model of FKRPs function we propose. This new confirmatory data, requested by the reviewer, is presented in Extended Data Figure 9a, b.

Q2.2. *The authors claimed myosin10 localized in the luminal side of Golgi. However, it could not conclude by STED study. Actually, location of Golgi(58k) is reported in the cytosol side of Golgi. They should give concrete data of location myosin10 by different methods.*

Response. We are happy to provide the further experiments requested by the reviewer. To undertake these additional analyses, we employed Airyscan super-resolution imaging in conjunction with a different Golgi marker to better localise Myosin10. The Airyscanner approach has the advantage of being able to 3D render the Golgi which allows us to better visualise localisation of Myosin10 as requested by the reviewer. Although we acknowledge that the Golgi 58K protein does localise on the cytosolic side of the Golgi, we wish to point out that it is a widely used general marker of the Golgi membrane. However, to broaden our analyses, in the manner the reviewer requests, we added an additional Golgi marker in the Airyscanner observations, Golgi Reassembly Stacking Protein 2 (GORASP2)¹, which is a general Golgi membrane protein.

We have now comprehensively extended our investigation to include 3 additional experimental and 3 additional technical replicates, imaging a total of 9 areas for each of the 9 replicates with the Airyscanner method. We also include now an additional supplementary video (2) rendered in 3D software that gives a better view inside the Golgi in super resolution. A rendered panel figure and a sphericity analysis of the new data (which nicely documents the changes that occur within the Golgi of mutant cells) and an analysis of co-localization of the new data using the Golgi as a mask is also provided. The new statistically validated, data confirms our findings from the original STED data. Collectively, this data confirms that Fibronectin is co-localized with Myosin10 in the Golgi of control cells, a distribution that is significantly altered in FKRPs patient-derived cell lines.

1. Zhang X, Wang Y. The Golgi stacking protein GORASP2/GRASP55 serves as an energy sensor to promote autophagosome maturation under glucose starvation. *Autophagy*. 2018;14(9):1649-1651

Q2.3. *The authors concluded that FKRP did not catalyze fibronectin sialylation directly (Ex Fig.9). line 892, there is no description how to prepare enzymes? Does this FKRP protein have ribitol-P-transferase activity?*

Response. We apologise for the methods oversight we have now added the requested enzyme preparation protocol into the manuscript methods at page 50. On the matter of testing if our FKRP protein has ribitol-P-transferase activity we do appreciate the reviewer's point here. Unfortunately, however, the reviewer's request to confirm that our FKRP protein has ribitol-P-transferase activity is a very difficult one to achieve. There are several technical challenges to establish a FKRP activity assay. Firstly, the known FKRP donor substrate CDP-ribitol, unlike CMP-Neu5Ac for sialylation, is not commercially available. Secondly, the enzyme activity required to add ribitol-P also requires the production of human ISPD (isoprenoid synthase domain-containing protein) to allow the *in vitro* synthesis of CDP-ribitol¹. Thirdly, the CDP-ribitol is added to a phosphorylated tetrasaccharide structure on α -DG. To synthesise this substrate requires production of the α -DG fragment and five enzymes (POMT1/2, SGK196, POMGNT2, B3GALNT2, and fukutin)². Apart from recombinant protein expression and production of all these enzymes, it would also require HPLC method development to purify products for each synthesis step. The complexity of this is further demonstrated in a recent publication by Kuwabara *et al.*³ where they further demonstrate that the core M3 Man-6-P peptide with Phosphate on the O-Mannose is necessary for enzymatic activity using Chromatography and MALDI-mass spectrometry. Thus, to perform this alternate assay in our context would require chemical synthesis of CDP Ribitol, isolation of Core M3 Phospho glycopeptide and further extensive analytical methods as shown by Kuwabara *et al.*, 2020 (Methods) and Fig 5, Suppl Fig 5 and Suppl Fig 6 (Results). In fact, the demonstration of this activity is a full manuscript in its own right and I hope the reviewer may concede that it is outside the scope of this study given that sialylation activity is the point under contention for this manuscript.

However, we have confirmed that the FKRP recombinant protein that we have produced is correct by both molecular mass and mass spectroscopic identification of FKRP via SDS-PAGE gel band extraction, which is how most commercially available recombinant proteins are assessed. We provide this data on the production of the protein in Extended Data Figure 9.

1. Gerin, I., Ury, B., Breloy, I. *et al.* ISPD produces CDP-ribitol used by FKTN and FKRP to transfer ribitol phosphate onto α -dystroglycan. *Nat Commun* **7**, 11534 (2016).
2. Kanagawa, M. *et al.* Identification of a Post-translational Modification with Ribitol-Phosphate and Its Defect in Muscular Dystrophy. *Cell Rep* **14**, 2209-2223, doi:10.1016/j.celrep.2016.02.017 (2016).
3. 2. Kuwabara, N. *et al.* Crystal structures of fukutin-related protein (FKRP), a ribitol-phosphate transferase related to muscular dystrophy. *Nat Commun* **11**, 303, doi:10.1038/s41467-019-14220-z (2020).

Q2.4. *It is of interest that the authors found the increment of ST6Gal1 expression in FKRP mutants although they tried to find down regulated genes. However, it is unclear why they could not find any difference total sialylation (Ex Fig. 6) in control and patients although ST6Gal1 increased?*

Response. The reviewer raises an interesting point here that had also been unclear to us. Prompted by the reviewer's comments we revisited this issue. In the original sample preparation protocol we performed for this analysis, we analysed the glycans released from the proteins in the cell lysate after the immunoprecipitation of fibronectin. We were concerned that this may have affected the sensitivity of the assay. We therefore repeated our analysis

using a different sample preparation method in which glycans were directly released from the membrane proteins of total cell lysates. This has indeed proved to be a better approach, which produced higher sensitivity and improved spectra from which to calculate the total cell protein sialylation. The total glycosylation now matches the RNA seq data as expected in that we see an increase in relative abundance of sialylation, specifically in 2,6 sialylation in the patient samples that correlates with the ST6Gal1 increase in expression. The data is provided in a revised Extended Data Figure 6.

Q2.5. *Lines 824, 825, pathological information including severity of CMD patient and LGMD2I patient was not described at all. They may be different. IIH6 staining patterns of both should be shown.*

However, sialylation of both fibronectin did not show any difference (Fig. 3b), suggesting that sialylation level of fibronectin does not correlate directly with disease severity. By the way, in Fig. 3a, Red box may be Blue box. Line 479, g should be removed. It is noted that this review could not see the increase of asialobianttenary N-glycan peak in patients. Why?

Response. We agree with the reviewer that more clinical data is an important addition to the manuscript. We have now included a table (Table 1) detailing clinical phenotype of the patients from which cells were derived. We also include information on both the IIH6 and VIA4 status on sections and referenced publications concerning these patients.

In terms of the same sialylation levels of fibronectin in both patient samples we tackle this question in response to Q1 of Reviewer1 and refer the Reviewer 2 to this answer.

The red box is indeed blue and we have now altered the corresponding text. We apologize for this error.

To answer the query as to whether there is a complementary increase of asialylated species with the decrease in sialylation observed in the patients we undertook further analyses of our spectra. As we reported in the original submission the blue boxed peaks m/z 965.9 and 1111.4 in Fig 3a, that represent mono- and di- sialylated biantennary glycans in the control, are lost in the patient samples and the reviewer is correct, that this should lead to a corresponding increase in asialobianttenary N-glycans. Prompted by the reviewer's insightful question we have now revisited our data and we do see the corresponding increase of asialo N-glycans on fibronectin isolated from the mutant cells, relative to the sialylated species (as shown in the following figure extracted from Fig 3a). It confirms the loss of sialylation is accompanied by increase of asialo-species. We provide the data here as we think as it essentially duplicates Fig 3a, but are happy to include it in the manuscript if the reviewer thinks it is a necessary point of emphasis in resubmission.

We can also see a distinct drop in asialobiantennary non-fucosylated (A2G2- m/z 820.39) and fucosylated (F1A2G2- m/z 893.32) peaks in the total cell lysate (Extended Data Fig.6) in the FKR mutant patient cell lines compared to healthy controls. Accordingly, a complementary increase in relative abundance of asialylated species relative to the sialylated species is observed.

Q2.6. Line 240, CDP-ribitol should be ribitol-phosphate.

Response. The text has been altered as requested.

Q2.7. Lines 553~555, this explanation is wrong, because fukutin and FKRPs sequentially contributed to the α -dystroglycan-glycan formation.

Response. This explanation has been altered to read “Levels of glycosylated α -dystroglycan, which are dependent on FKRPs activity were assessed using IIIH6 antibody immunoreactivity.”

Q2.8. Two reference sections could be found (page 12 and page 48). The latter one may be correct.

Response. The two reference lists (main text and methods) have been integrated in line with specific NComms format. We thank the reviewer for pointing out this oversight.

Q2.9. Fig. 4, why fibronectin staining patterns of patients are different from control?

Response: As outlined in response to Q2.2 the staining patterns of the fibronectin within patient-derived cells is different to control because its localization is altered within the Golgi when it is not fully processed. We provide additional data using the Airy scanner in Extended figures 4e-g and Supplementary Video 2 to further document and illustrate this phenomenon, which is a specific finding of our paper.

Q2.10. Table 1, why the authors did not examine myosin 9? Refs 32 and 33 are related to myosin 9 (non-muscle myosin IIA).

Response: We agree with the reviewer that Myosin9 also presents as a possible FKRPs interacting partner based on the proteomic screen and refs 32-33. However, important work from the Stainier group¹, which we cite in the submission, indicated that *Myosin10* deficiency led to very similar tissue elasticity deficits in the respiratory system of mutant mice to those we document in the myotome of our *fkrp* mutants. Based on this information we chose to focus on the role of Myosin10, given it had already been functionally implicated in a similar process in another tissue. We note, however, that our results do not preclude the possibility of a role for Myosin9 in mediating FKRPs function and note that the identification of these two highly similar proteins within our proteomic screen is reassuring of the veracity of the binding results for this class of proteins to FKRPs.

1. Kim, H. T. *et al.* Myh10 deficiency leads to defective extracellular matrix remodeling and pulmonary disease. *Nat Commun* **9**, 4600, doi:10.1038/s41467-018-06833-7 (2018).

Q2.11. *Ex Fig. 1, i, j and k are missing.*

Response: Labels added as requested.

Q2.12. *Ex Fig. 2c'', Laminin intensity data should be provided instead of FWHM.*

Response: We have previously found¹ full width half maximal analysis (FWHM) to be a more accurate way to analyze laminin in the septa since the polyclonal pan laminin antibody we use is currently the only known laminin antibody that works well in fish. It is known to detect laminin-111, laminin-211 and laminin-411 therefore it detects a range of laminins across the septal basement membranes and does not have the defined detection of the 43Dag antibody and IIH6. With a broader expression, the local image analysis experts (Monash Micro Imaging) suggested the FWHM would be a better approach to assess laminin at the septa. We have previously used this analysis approach in our recent HMG publication:

1. Wood, A. J. *et al.* RGD inhibition of itgb1 ameliorates laminin-alpha2-deficient zebrafish fibre pathology. *Hum Mol Genet* **28**, 1403-1413, doi:10.1093/hmg/ddy426 (2019).

Q2.13. *Ex Fig. 7 legend is insufficient description. This suggests sialylation-independent binding of fibronectin-collagen also contributed?*

Response: We apologize for the confusion in this figure legend. We have altered the text to better describe the experiment in line with the review comments. Extended Data Figure 7 is essentially a control experiment for Fig. 3d. In this experiment, the neuraminidase enzyme is not altered in concentration, it is the concentration of fibronectin that is labelled on the graph. Fibronectin is initially bound to the chip and after a standard treatment of neuraminidase, collagen is then run over the chip and binding levels determined. This experiment is an essential control to demonstrate the dose response of fibronectin is not altered as collagen binding increases. Importantly the supplementary data shows that the shape of the curve does not change with the dose of fibronectin, this is a critical control for this type of analysis as many Biacore experiments only work at an optimal dose. In our case the dose of fibronectin does not alter the binding kinetics providing strong evidence the effect is robustly reproducible.

Q2.14. *First appearance of MBM (line 75). There is no explanation.*

Response. The text has been altered as requested.

Q2.15. *There are many type-errors and reference numbers (e.g. line 673, ref 439). These hampered the review process.*

Response: The text has been altered at requested and the manuscript reviewed for typographical errors.

Reviewers' Comments:

Reviewer #1:

Remarks to the Author:

The authors appear to have made significant edits to the manuscript since the previous submission. I appreciate the improvements including the Myosin10-FKRP co-IP interaction that myself and the other reviewer raised. The major claims are now fully supported. I also appreciate the extended vascular and retina phenotyping of the fkrp mutant fish in extended figures 2 and 3. I think this is essential to accurately report any differences and discrepancies between the loss-of-function fkrp mutants and the fkrp morphants. The overall labeling and terminology has also been cleaned up. Statistics are appropriate when used. I have no other issues and I believe that this manuscript will be a good addition to the field of FKRP and CMD/LGMD biology.

Reviewer #2:

Remarks to the Author:

Revisions made great improvements and new added data strengthens the authors' claims. However, still some editorial problems could be found. (For example. Lines 1, 47, 313, Fibronectin is fibronectin; line 48, the two, major, two commas could be removed; line 170, maybe is may be; line 307, only the first letter is uppercase; Ref 50 and Ref 54, journal should be abbreviated, and Ref 54, first names of authors could be removed; etc).

REVIEWERS' COMMENTS

Reviewer 1.

The authors appear to have made significant edits to the manuscript since the previous submission. I appreciate the improvements including the Myosin10-FKRP co-IP interaction that myself and the other reviewer raised. The major claims are now fully supported. I also appreciate the extended vascular and retina phenotyping of the fkrp mutant fish in extended figures 2 and 3. I think this is essential to accurately report any differences and discrepancies between the loss-of-function fkrp mutants and the fkrp morphants. The overall labeling and terminology has also been cleaned up. Statistics are appropriate when used. I have no other issues and I believe that this manuscript will be a good addition to the field of FKRP and CMD/LGMD biology.

Response: The reviewer appreciates the extent we have gone to in review to address their questions and raises no further questions to address.

Reviewer 2.

Revisions made great improvements and new added data strengthens the authors' claims. However, still some editorial problems could be found. (For example. Lines 1, 47, 313, Fibronectin is fibronectin; line 48, the two, major, two commas could be removed; line 170, maybe is may be; line 307, only the first letter is uppercase; Ref 50 and Ref 54, journal should be abbreviated, and Ref 54, first names of authors could be removed; etc).

Response. Reviewer2 also appreciates the extent of the revisions we have undertaken and has no further experimental concerns. She/he does raise some individual typographical issues all of which we have individually addressed. The manuscript and figures have again been proof read and typographical errors we have found have been corrected. All the changes are marked in purple in the manuscript.